

# Differential effects of cisplatin on cybrid cells with varying mitochondrial DNA haplogroups

Sina Abedi[1], Gregory Yung[1], Shari R. Atilano[1], Kunal Thaker[1], Steven Chang[1], Marilyn Chwa[1], Kevin Schneider[1], Nitin Udar[1], Daniela Bota[2] and M. Cristina Kenney[1,3]

[1] Gavin Herbert Eye Institute, University of California, Irvine, Irvine, CA, United States of America
[2] Department of Neurology, Neuro-Oncology Division, University of California, Irvine CA, United States of America
[3] Department of Pathology and Laboratory Medicine, University of California, Irvine, CA, United States of America

Corresponding authors
Sina Abedi, sabedi1@uci.edu
M. Cristina Kenney, mkenney@hs.uci.edu

## ABSTRACT

**Background**. Drug therapy yields different results depending on its recipient population. Cisplatin, a commonly used chemotherapeutic agent, causes different levels of resistance and side effects for different patients, but the mechanism(s) are presently unknown. It has been assumed that this variation is a consequence of differences in nuclear (n) DNA, epigenetics, or some external factor(s). There is accumulating evidence that an individual's mitochondrial (mt) DNA may play a role in their response to medications. Variations within mtDNA can be observed, and an individual's mtDNA can be categorized into haplogroups that are defined by accumulations of single nucleotide polymorphisms (SNPs) representing different ethnic populations.
**Methods**. The present study was conducted on transmitochondrial cytoplasmic hybrids (cybrids) that possess different maternal-origin haplogroup mtDNA from African (L), Hispanic [A+B], or Asian (D) backgrounds. Cybrids were created by fusing Rho0 ARPE-19 cells (lacking mtDNA) with platelets, which contain numerous mitochondria but no nuclei. These cybrid cells were cultured to passage five, treated with cisplatin, incubated for 48 h, then analyzed for cell metabolic activity (tetrazolium dye (MTT) assay), mitochondrial membrane potential (JC-1 assay), cytotoxicity (lactate dehydrogenase (LDH) assay), and gene expression levels for *ALK*, *BRCA1*, *EGFR*, and *ERBB2/HER2*.
**Results**. Results indicated that untreated cybrids with varying mtDNA haplogroups had similar relative metabolic activity before cisplatin treatment. When treated with cisplatin, (1) the decline in metabolic activity was greatest in L (27.4%, $p < 0.012$) < D (24.86%, $p = 0.0001$) and [A+B] cybrids (24.67%, $p = 0.0285$) compared to untreated cybrids; (2) mitochondrial membrane potential remained unchanged in all cybrids (3) LDH production varied between cybrids (L > [A+B], $p = 0.0270$). (4) The expression levels decreased for *ALK* in L ($p < 0.0001$) and [A+B] ($p = 0.0001$) cybrids but not in D cybrids ($p = 0.285$); and decreased for *EGFR* in [A+B] cybrids ($p = 0.0246$) compared to untreated cybrids.
**Conclusion**. Our findings suggest that an individual's mtDNA background may be associated with variations in their response to cisplatin treatment, thereby affecting the efficiency and the severity of side effects from the treatment.

## INTRODUCTION

Cisplatin is a platinum complex used in the chemotherapeutic treatments of various cancers, including ovarian and testicular cancers. It functions primarily by forming intrastrand crosslink adducts with the cell's DNA. This activates signal transduction pathways that disrupt transcription and replication, therefore inducing apoptosis in target cells (*Dasari & Tchounwou, 2014*; *Florea & Büsselberg, 2011*; *Meijera et al., 2001*; *Siddik, 2003*). Side effects of cisplatin treatment include nausea, vomiting, hearing loss, myelosuppression, neurotoxicity, nephrotoxicity, hepatotoxicity, and cardiotoxicity (*Dasari & Tchounwou, 2014*). Unfortunately, cisplatin resistance is common, and the biochemical mechanisms leading to resistance are only partially understood (*Amable, 2016*; *Boulikas & Vougiouka, 2004*; *Cocetta, Ragazzi & Montopoli, 2019*; *Damia & Broggini, 2019*; *Florea & Büsselberg, 2011*; *Gottesman et al., 2016*; *Siddik, 2003*). Studies have shown that mechanisms related to DNA repair, decreased drug uptake, or decreased DNA adduct formation have been associated with increased cisplatin resistance (*Amable, 2016*; *Damia & Broggini, 2019*; *Siddik, 2003*).

Variations in chemosensitivity to cisplatin have been observed in individuals of varying ethnic backgrounds (*Jing, Su & Ring, 2014*; *Kenney et al., 2014b*; *O'Donnell & Dolan, 2009*; *Tan, Mok & Rebbeck, 2016*; *Yasuda, Zhang & Huang, 2008*). In a study investigating genetic biomarkers, Caucasian individuals demonstrated significantly greater susceptibility to identical doses of cisplatin compared to Asian individuals with late-stage non-small lung cancer (NSCLC) (*Rose, Kostyanovskaya & Huang, 2014*). *O'Donnell & Dolan (2009)* found that individuals of Asian descent (excluding India) were more susceptible to chemotherapy agents than individuals of European descent, and additionally, individuals of African descent showed greater resistance to anticancer treatments than those of European descent. These differences have previously been attributed to differences within the nuclear DNA and epigenetic factors (*Amable, 2016*; *Ha et al., 2018*; *Housman et al., 2014*; *O'Byrne, Barr & Gray, 2011*; *Siddik, 2003*; *Stewart, 2007*; *Ziliak et al., 2012*).

Currently, evidence demonstrates that the mitochondria may also affect an individual's response to cisplatin treatment (*Han et al., 2017*; *Marrache, Pathak & Dhar, 2014*; *Marullo et al., 2013*; *Tsuyoshi et al., 2017*). Mitochondria are organelles that play an important role in ATP production, calcium homeostasis, cell signaling, and apoptosis. Previous studies have indicated that cisplatin treatment results in mitochondrial damage leading to reduced energy production and increased apoptotic activity (*Devarajan et al., 2002*; *Han et al., 2017*; *Park, De Leon & Devarajan, 2002*; *Patel et al., 2019*; *Tsuyoshi et al., 2017*; *Yáñez et al., 2003*). Mitochondria are unique in that they contain their own DNA. Mitochondrial (mt) DNA is maternally inherited and can be categorized into haplogroups that are defined by single nucleotide polymorphism (SNP) variants. Different mtDNA haplogroups are used to classify populations of different maternal ancestral origins (http://www.mitomap.org/

foswiki/pub/MITOMAP/MitomapFigures/WorldMigrations2012.pdf). Over the span of thousands of years, the different ancestral populations have accumulated distinct patterns of SNP variants within the mitochondrial genome, and it is likely that these SNP variants contribute to changes in cellular functions (*Atilano et al., 2015*; *Jing, Su & Ring, 2014*; *Kenney et al., 2014a*; *Kenney et al., 2014b*; *Kenney et al., 2013*; *Marrache, Pathak & Dhar, 2014*; *O'Donnell & Dolan, 2009*; *Patel et al., 2019*; *Penta et al., 2001*; *Singh et al., 1999*; *Tan, Mok & Rebbeck, 2016*; *Van Gisbergen et al., 2015*).

Damage to the mtDNA accumulates because mitochondria, unlike the nucleus, lack efficient DNA repair mechanisms. Germline and somatic mutations, as well as copy number differences within the mtDNA, are associated with increased cancer risk. These mutations are likely to remain within the genome and be passed on to offspring (*Van Gisbergen et al., 2015*). Cisplatin susceptibility within the mitochondria could, however, be lessened. Variations within the mtDNA that prevent cisplatin from affecting ATP production, altering signaling pathways, or forming DNA adducts could lead to cisplatin resistance (*Czarnecka & Bartnik, 2011*; *Florea & Büsselberg, 2011*; *Jarrett et al., 2008*; *Kenney et al., 2014b*; *Kenney et al., 2013*; *Singh et al., 1999*). Similarly, in a study on pancreatic cancer cells, Mizutani et al. demonstrated that mutations within the mitochondrial genome could result in cell resistance to cisplatin (*Mizutani et al., 2009*).

In order to study the effect(s) of differing mtDNA variants on drug susceptibility, the cybrid model can be used. Cybrids are transmitochondrial cell lines that contain identical nuclei, but mtDNA from different patients. These cybrids vary only in their mtDNA; therefore, differences in cellular function(s) between cybrids of varying mtDNA haplogroups can be attributed to their different mtDNA. Previously published data within our laboratory illustrated that transmitochondrial cybrids with mitochondria from either mtDNA haplogroup H (Southern Europe) or J (Northern Europe), exhibited varying levels of cellular metabolism, cell viability, and gene expression levels in vitro. J mtDNA haplogroup cybrids showed metabolism levels similar to those of cancer cells, as defined by the *Warburg Phenomenon* (*Kenney et al., 2014a*; *Kenney et al., 2013*; *Patel et al., 2019*).

We designed the present study to observe the effect(s) that mtDNA from L (African maternal descent), [A+B] (Hispanic maternal descent), and D (Asian maternal descent) haplogroups have on cellular function(s) after cisplatin treatment. Our findings demonstrated that treating cybrids containing varying mtDNA haplogroups with cisplatin yielded differing levels of cellular metabolic activity, cytotoxicity, and gene expression. These findings support the hypothesis that mtDNA representing different racial/ethnic groups likely plays a role in the effectiveness and side effects induced by cisplatin.

## MATERIALS & METHODS

### Creation of cybrid cells

Cytoplasmic hybrids (cybrids), cell lines with identical nuclear genomes but different mitochondrial genomes, were created and used in the fifth passage for all experiments. All experiments were carried out in accordance with the Institutional Review Board at the University of California, Irvine, (IRB #2003-3131) and were consistent with Federal

guidelines. All subjects in this study read and signed the written informed consent prior to beginning their participation. Cybrid cells were created using platelets isolated from peripheral blood that was fused with Rho0 (mtDNA free) ARPE-19 cells (Fig. 1) (*Patel et al., 2019*). The mtDNA haplogroups for each subject and cybrid cell line were identified using polymerase chain reaction (PCR) along with restriction enzyme digestion and mtDNA sequencing as described previously in the study conducted by *Patel et al. (2019)* on H and J mtDNA haplogroup cybrid cells.

## Sequencing of mtDNA from L, A, B, and D Cybrids

DNA was extracted from the individual cybrids ($n = 7$ for L cybrids, $n = 4$ for [A+B] cybrids, and $n = 3$ for D cybrids) using a kit (DNeasy Blood and Tissue Kit, Qiagen, Germantown, MD). Sequencing techniques similar to those previously described in the study between H and J mtDNA haplogroup cybrids were conducted (*Patel et al., 2019*). Next Generation Sequencing (NGS) technology was used to sequence both strands of mtDNA independently in both directions. This was done to quantitate the haplogroup-defining single nucleotide polymorphisms (SNPs), private SNPs (not defining haplogroups), and low-frequency heteroplasmy SNPs across the entire mitochondrial genome. NGS technology is capable of deep sequencing (average sequencing depth of 30,000; range 1,000 to 100,000) and accurately differentiates low-frequency mtDNA heteroplasmy SNPs from DNA modification artifacts. The mtDNA sequences were analyzed using HaploGrep (https://haplogrep.i-med.ac.at/) to identify the mtDNA haplogroups (Figs. 2–4, Table 1). Defined SNP changes between the L, [A+B], and D cybrids were obtained using http://www.Phylotree.org. and amino acid changes and any associated pathogenesis resulting from differences in SNP variants were verified using http://www.MitoMap.org and/or http://www.hmtvar.uniba.it. The rs numbers were identified using http://www.ncbi.nlm.nih.gov/snp. All of the SNPs identified had a Quality Score of 100 (A Phred-scaled quality score assigned by the variant caller) and Passed all of the Filters (*Patel et al., 2019*) (Table 2).

## Cybrid cell lines culture

Cybrid cell lines containing mitochondria with either L, [A+B], or D haplogroups were cultured in flasks until confluent. They were then plated on a 96-well or 24-well plates depending on the assay. The MTT, JC1 and LDH experiments were performed on 7 different L cybrids, each with mitochondria from different individuals. The [A+B] cybrids represented 4 different individuals. The D cybrids represented 3 different subjects. Each treatment condition was run in quadruplicate and the experiments repeated twice. The mean values of all cybrids were calculated for both experiments individually. Then these individual means were combined by haplogroup (L versus [A+B] versus D) for the final analyses.

Cells were plated in 96-well plates with each well containing 50,000 cells. After 24 h, media were removed, and cells were treated with 100 μL of 40 μM concentration of cisplatin (given as a gift from Professor Bota, UCI Medical Director, Neuro-Oncology Program). Generally, the concentration of cisplatin used to treat cancer cells for 48 h is 10 μM.
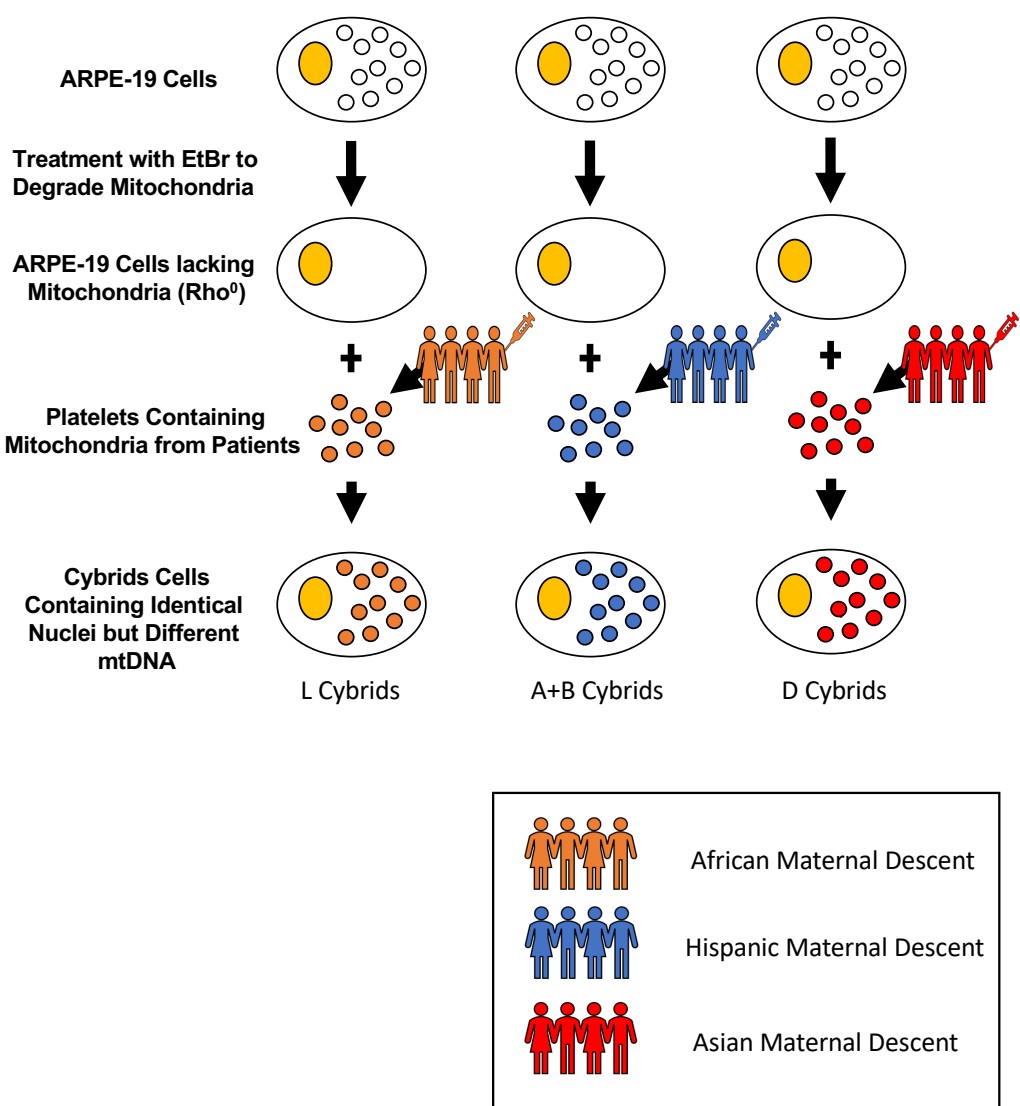

**Figure 1  Creation of cybrid cells with identical nuclei but varying mtDNA.** Cytoplasmic hybrids (cybrids) were created by fusing ARPE-19 Rho0 (mtDNA free) cells with platelets isolated from peripheral blood from individuals of either L (African), [A+B] (Hispanic), or D (Asian) maternal descent. Rho0cells were initially obtained by treating ARPE-19 cells with small doses of Ethidium Bromide (EtBr) over many passages until they lacked mitochondria. Once the cells were depleted of their mitochondria, they were fused with the obtained platelets using polyethylene glycol fusion. These platelets lacked nuclear (n) DNA, and only contained mitochondrial (mt) DNA. Once fused, the created cybrids contained identical nDNA and the particular mtDNA obtained from the specified patient's platelets. The mtDNA haplogroups for each subject and cybrid cell line were identified using polymerase chain reaction (PCR) along with restriction enzyme digestion and mtDNA sequencing.

However, a previous study done by Patel et al. on ARPE-19 cybrids containing either H or J mtDNA haplogroups found that the concentration best fit to inhibit these cells was 40 μM. This was concluded via IC-50 analysis which demonstrates the concentration of cisplatin necessary to inhibit cell viability by 50%. As a result, the 40 μM concentration was
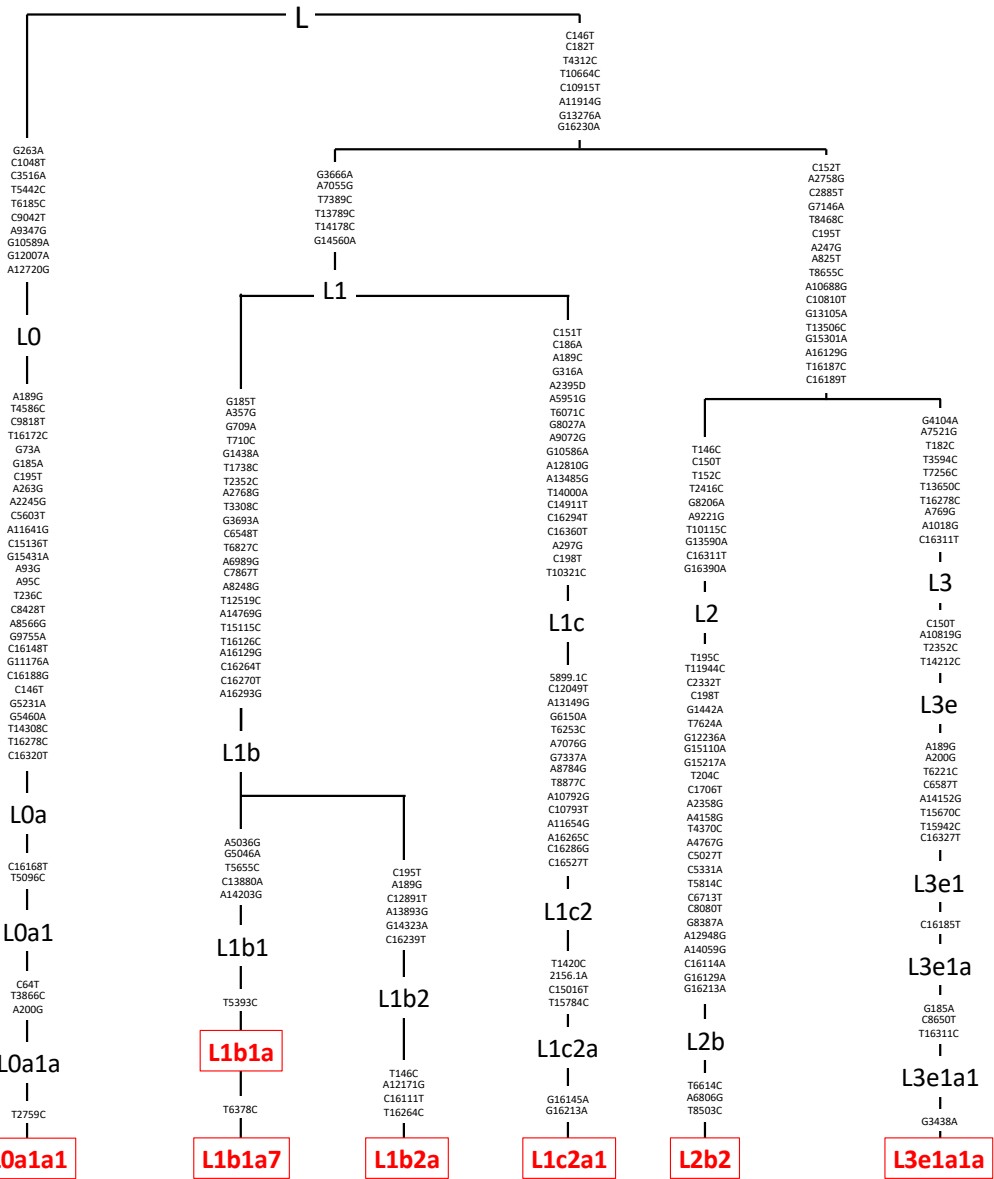

**Figure 2** **Mitochondrial SNP variants associated with the various L mtDNA haplogroup subtypes used in this study.** Platelets obtained from patients were sequenced using a kit (DNeasy Blood and Tissue Kit, Qiagen, Germantown, MD), and found to be in the L mtDNA haplogroup ($n = 7$). The L mtDNA haplogroup is associated with individuals from African maternal descent. The specific variants of those in the L mtDNA haplogroup were identified and their phylogenic tree is displayed. These variants include L0a1a1, L1b1a, L1b1a7, L1b2a, L1c2a1, L2b2, and L3e1a1a. Next Generation Sequencing (NGS) technology was used to sequence both strands of mtDNA independently in both directions. This was done to quantitate the haplogroup-defining single nucleotide polymorphisms (SNPs), private SNPs (not defining haplogroups), and low-frequency heteroplasmy SNPs across the entire mitochondrial genome. The mtDNA sequences were analyzed using HaploGrep (https://haplogrep.i-med.ac.at/) to identify the mtDNA haplogroups. SNP variants were verified using http://www.MitoMap.org, http://www.Phylotree.org, and/or http://www.hmtvar.uniba.it.

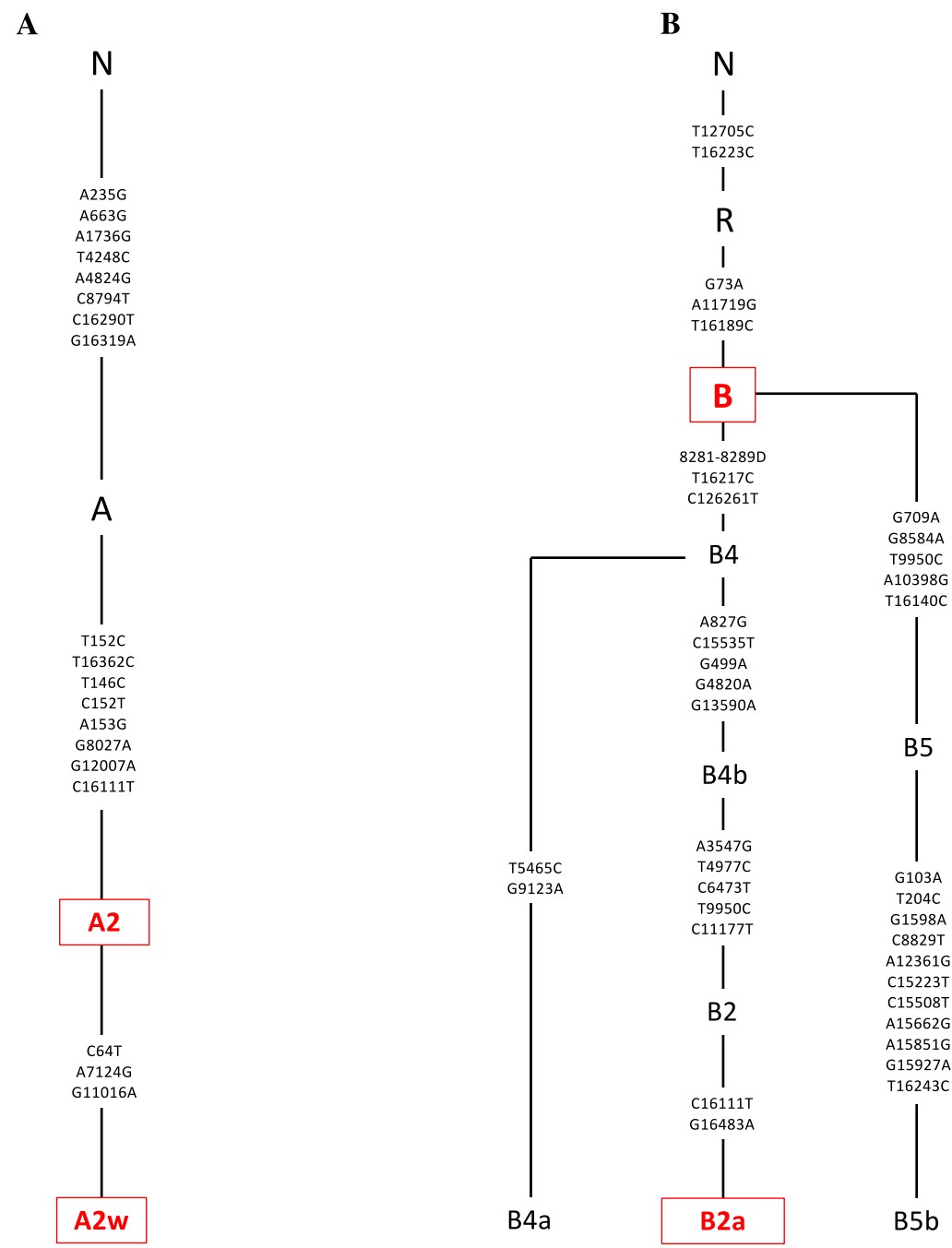

**Figure 3 Mitochondrial SNP variants associated with the A and B mtDNA haplogroups used in this study.** Platelets obtained from patients were sequenced using a kit (DNeasy Blood and Tissue Kit, Qiagen, Germantown, MD), and found to be in the A or B mtDNA haplogroups ($n = 4$). (A) The A mtDNA haplogroup is associated with individuals from Hispanic maternal descent. The specific variants of those in the A mtDNA haplogroup were identified and their phylogenic tree is displayed. These variants include A2 and A2w. (B) The B mtDNA haplogroup is also associated with individuals from Hispanic maternal (continued on next page…)

**Figure 3 (…continued)**
descent. The specific variants of those in the B mtDNA haplogroup were identified and their phylogenic tree is displayed. These variants include B and B2a. Next Generation Sequencing (NGS) technology was used to sequence both strands of mtDNA independently in both directions. This was done to quantitate the haplogroup-defining single nucleotide polymorphisms (SNPs), private SNPs (not defining haplogroups), and low-frequency heteroplasmy SNPs across the entire mitochondrial genome. The mtDNA sequences were analyzed using HaploGrep (https://haplogrep.i-med.ac.at/) to identify the mtDNA haplogroups. SNP variants were verified using http://www.MitoMap.org, http://www.Phylotree.org, and/or http://www.hmtvar.uniba.it.

selected for treatment in the current study (*Patel et al., 2019*). Plates were then incubated at 37 °C for another 48 h.

## Cellular metabolic activity measured by tetrazolium dye (MTT) assay

The MTT assay was conducted to assess cellular metabolic activity, which was in turn correlated to ATP production and overall cell viability. After the 48-hour incubation period, 10 μL of MTT solution (3-(4,5-dimethylthiazol-2-yl)-2,5-diphenyltetrazolium bromide, (Biotium, Hayward, CA) was added to the 100 μL of medium in each well. Plates were then incubated at 37 °C for 2 h. 100 μL of DMSO was directly added into the medium in each well and pipetted up and down until thoroughly mixed. Absorbance was measured on an absorbance reader (BioTek, ELX 808, Winooski, VT) at 570 nm (measured) and 630 nm (reference). The reference absorbance value was subtracted from the measured value in order to attain true absorbance values. All values were normalized to the average metabolic activity of the untreated-L cybrids. Average values for the cybrids were then compared to the respective untreated cybrids using a two-tailed $t$-test (GraphPad Prism Software, Inc, San Diego, CA). $P \leq 0.05$ was deemed statistically significant. Each treatment condition was run in quadruplicate, and the entire experiment was repeated twice.

## Mitochondrial membrane potential (ΔΨm) assay (JC-1 assay)

L, [A+B], and D cybrids were plated in 24-well plates (100,000 cells/well), incubated 24 h and, treated with 0 or 40 μM of cisplatin for another 48 h. JC-1 reagent (5, 5′, 6, 6′-tetrachloro-1,1′, 3, 3′- tetraethylbenzimidazolylcarbocyanine iodide) (Biotium, Hayward, CA) was added to cultures for 15 min. Similar to the study conducted by Patel et al. a Gemini XPS Microplate Reader (Molecular Devices) was used to measure fluorescence for red (excitation 550 nm and emission 600 nm) and green (excitation 485 nm and emission 545 mm) wavelengths. Intact mitochondria with normal ΔΨm fluoresced red, while cells with decreased ΔΨm fluoresced green. Experiments were then analyzed in quadruplicate, and the entire experiment was repeated twice. All values were normalized to the average of the untreated-L cybrids, and cisplatin-treated cybrids were compared to untreated cybrids using a two-tailed $t$-test to assess for statistical significance (GraphPad Prism Software, Inc.) (*Patel et al., 2019*).

## Mitochondrial membrane potential (ΔΨm) assay (IncuCyte® live cell analyzer with ARPE-19 cells)

After cisplatin treatment, the cybrids showed decreased metabolic activity but the mitochondrial membrane potentials remained unchanged, which was unexpected.

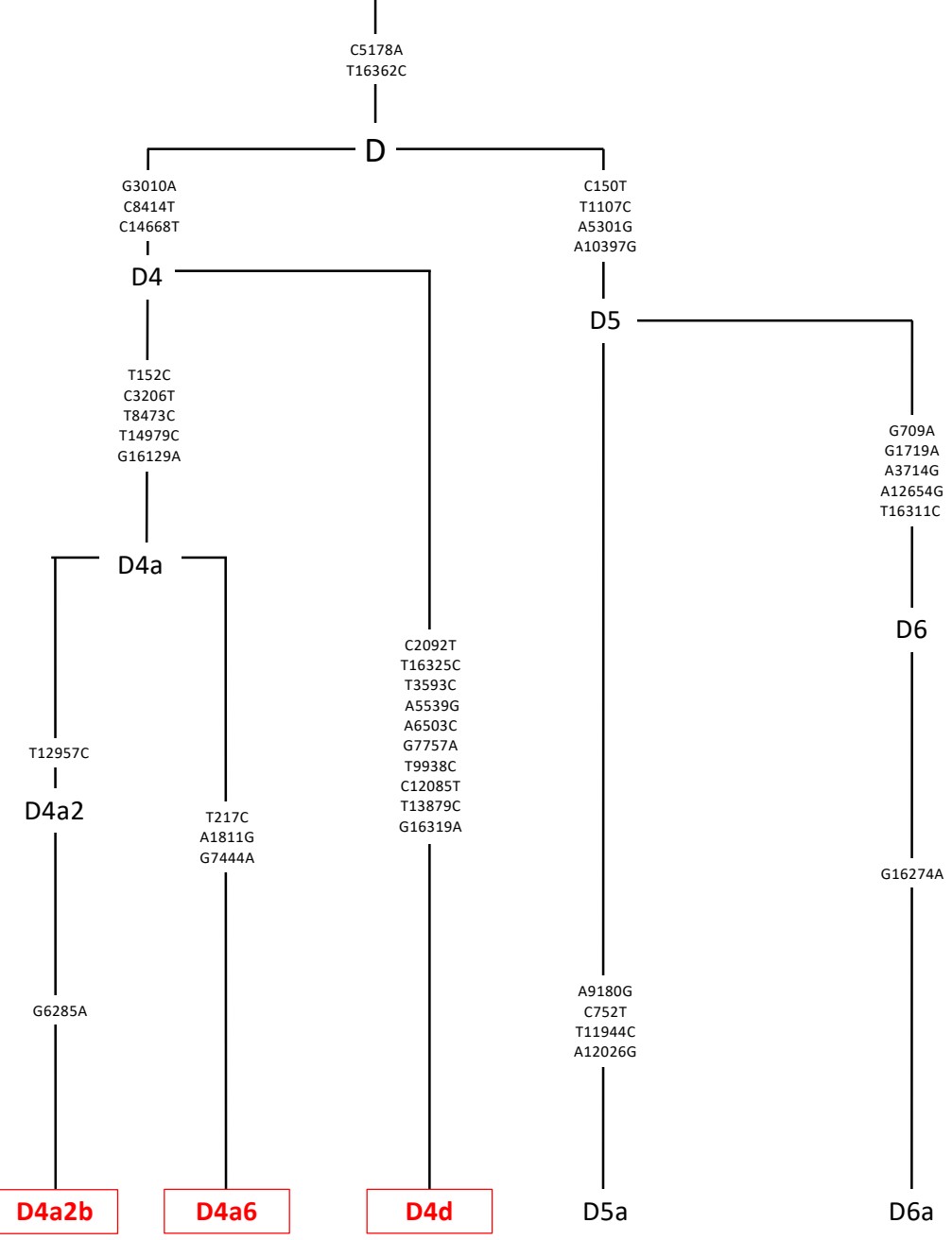

**Figure 4** **Mitochondrial SNP variants associated with the specific D mtDNA haplogroup types used in this study.** Platelets obtained from patients were sequenced using a kit (DNeasy Blood and Tissue Kit, Qiagen, Germantown, MD), and found to be in the D mtDNA haplogroup ($n = 3$). The D mtDNA haplogroup is associated with individuals from Asian maternal descent. The specific variants of those in the D mtDNA haplogroup were identified and their phylogenic tree is displayed. These variants include D4a2b, D4a6, and D4d. Next Generation Sequencing (NGS) technology was used to sequence both strands of mtDNA independently in both directions. (continued on next page...)

**Table 1** **Haplogroup profiles for the ARPE-19 cybrids.** Illustration of cybrid cell lines used with their associated haplogroup type along with the age and gender of the respective donor. Cytoplasmic hybrids (cybrids), cell lines with identical nuclear genomes but different mitochondrial genomes, were created and used in the fifth passage for all experiments. Cybrid cells were created using platelets isolated from peripheral blood that was fused with Rho0 (mtDNA free) ARPE-19 cells. The mtDNA haplogroups for each subject and cybrid cell line were identified using polymerase chain reaction (PCR) along with restriction enzyme digestion and mtDNA sequencing. DNA was extracted from the individual cybrids ($n = 7$ for L cybrids, $n = 4$ for [A+B] cybrids, and $n = 3$ for D cybrids) using a kit. Next Generation Sequencing (NGS) technology was used to sequence both strands of mtDNA independently in both directions. This was done to quantitate the haplogroup-defining single nucleotide polymorphisms (SNPs), private SNPs (not defining haplogroups), and low-frequency heteroplasmy SNPs across the entire mitochondrial genome. All of the SNPs identified had a Quality Score of 100 and Passed all of the Filters.

| Subject | Haplogroup | Age | Gender |
|---|---|---|---|
| Cyb L 11-38 | L0a1a1 | 38 | F |
| Cyb L 11-30 | L1b1a | 54 | F |
| Cyb L 13-126 | L1b1a7 | 39 | F |
| Cyb L 13-124 | L1b2a | 31 | M |
| Cyb L 13-125 | L1c2a1 | 52 | F |
| Cyb L 11-31 | L2b2 | 42 | M |
| Cyb L 11-17 | L3e1a1a | 64 | F |
| Cyb A 11-16 | A2 | 42 | F |
| Cyb A 11-14 | A2w | 34 | F |
| Cyb B 12-40 | B | 31 | M |
| Cyb B 13-68 | B2a | 45 | F |
| Cyb D 13-55 | D4a2b | 45 | F |
| Cyb D 11-18 | D4a6 | 39 | M |
| Cyb D 11-29 | D4d | 37 | F |

Therefore, we conducted additional experiments using the ARPE-19 cell line evaluated with the IncuCyte® Live Cell Analyzer (Sartorius, City, State) and the Nuclight Rapid Red probe (A549 NucLight Red, cat # 4491) that allowed for comparison of growth/proliferation over 40 h in cultures treated with 20 μM or 40 μM cisplatin ($n = 7$ per treatment condition). The IncuCyte® Analyzer captures changes in probes fluorescence and morphologic images in real time. The mitochondrial membrane potential was measured with the JC-1 reagent (5,5′,6,6′-tetrachloro-1,1′,3,3′- tetraethylbenzimidazolyl-carbocyanine iodide) (Biotium, Hayward, CA) that was added to cultures for 15 min. Fluorescence was measured for red (excitation 550 nm and emission 600 nm) and green (excitation 485 nm and emission 545

**Table 2 L, A+B, and D mtDNA haplogroup ARPE-19 cybrids displayed unique SNP variants.** SNP Variants associated with the L, [A+B], and D mtDNA haplogroups were compared to one another. Nucleotide changes containing one or more differences between the studied haplogroups were identified and analyzed. The location of the nucleotide change within the mtDNA, the base pair changed, their associated rs number, their presence in the L, [A+B], or D cybrids, and any associated pathogenesis found in the literature were identified and displayed. Non-synonymous changes present in some but not all haplogroups tested were found at m.4824 A > G, rs ND; m.5178 C > A, rs28357984; m.5442 T > C, rs3020601; m.7146 G > A, rs372136420; m.8701 G > A, rs2000975; m.8794 C > T , rs2298007; m.10398 G > A, rs2853826; and m.13105 G > A, rs2853501. Data was incorporated using http://www.hmtvar.uniba.it as well as http://www.phylotree.org.

| Loci: MT- | SNP | AA Change | rs# | Present in L Cybrids | Present in A+B Cybrids | Present in D Cybrids | Related pathogenesis |
|---|---|---|---|---|---|---|---|
| D-Loop | 152 C>T | Noncoding | rs117135796 | No | Yes | Yes | Predisposition to ataxia, bladder cancer, aging, hepatocellular carcinoma, predisposition to breast cancer, lung cancer, nasopharyngeal carcinoma, stomach cancer, liver cancer, ovarian carcinoma |
| D-Loop | 182 T>C | Noncoding | rs41473347 | No | Yes | Yes | Predisposition to ataxia, bladder cancer, lung cancer |
| D-Loop | 195 C>T | Noncoding | rs2857291 | No | Yes | Yes | Bipolar disease, predisposition to ataxia, bladder cancer, aging, breast cancer, lung cancer |
| D-Loop | 235 A>G | Noncoding | rs3937037 | No | Yes | No | Lung cancer |
| D-Loop | 247 A>G | Noncoding | rs41334645 | No | Yes | Yes | Predisposition to ataxia |
| D-Loop | 263 G>A | Noncoding | rs2853515 | Yes | No | No | Oncocytic adenomas, predisposition to ataxia, breast cancer, nasopharyngeal carcinoma |
| RNR1 | 663 A>G | rRNA | ND | No | Yes | No | Hearing loss |
| RNR1 | 769 A>G | rRNA | rs2853519 | No | Yes | Yes | N/A |
| RNR1 | 825 A>T | rRNA | rs2853520 | No | Yes | Yes | N/A |
| RNR1 | 1018 A>G | rRNA | rs2856982 | No | Yes | Yes | Heteroplasmic variant |
| RNR1 | 1048 C>T | rRNA | rs2000974 | Yes | No | No | Heteroplasmic variant, hearing loss |
| RNR2 | 1736 A>G | rRNA | rs193303006 | No | Yes | No | Heteroplasmic variant |
| RNR2 | 2758 A>G | rRNA | rs2856980 | No | Yes | Yes | N/A |
| RNR2 | 2885 C>T | rRNA | rs2854130 | No | Yes | Yes | N/A |
| ND1 | 3516 C>A | syn:L-L | rs2854132 | Yes | No | No | N/A |
| ND1 | 3594 T>C | syn:V-V | rs193303025 | No | Yes | Yes | Thyroid tumor |
| ND1 | 4104 G>A | syn:L-L | rs1117205 | No | Yes | Yes | N/A |
| ND1 | 4248 T>C | syn:I-I | rs9326618 | No | Yes | No | Homoplasmic variant |
| ND2 | 4824 A>G | **non-syn:T-A** | ND | **No** | **Yes** | **No** | Glaucoma |
| ND2 | 5178 C>A | **non-syn:L-M** | rs28357984 | **No** | **No** | **Yes** | Infertility, increased longevity, diabetes, atherosclerosis, hypertension, myocardial infarction, Parkinson's disease, glaucoma, blood iron metabolism, and cardiovascular disease |
| ND2 | 5442 T>C | **non-syn:F-L** | rs3020601 | **Yes** | **No** | **No** | Risk of mobility impairment |
| CO1 | 6185 T>C | syn:F-F | rs1029272 | Yes | No | No | Primary Open-Angle Glaucoma (POAG) |

**Table 2** (*continued*)

| Loci: MT- | SNP | AA Change | rs# | Present in L Cybrids | Present in A+B Cybrids | Present in D Cybrids | Related pathogenesis |
|---|---|---|---|---|---|---|---|
| CO1 | 7146 G>A | **non-syn:T-A** | rs372136420 | **No** | **Yes** | **Yes** | Asthenospermia |
| CO1 | 7256 T>C | syn:N-N | ND | No | Yes | Yes | N/A |
| TD | 7521 A>G | tRNA | rs200336937 | No | Yes | Yes | Thyroid cancer, colorectal tumor |
| ATP8 | 8468 T>C | syn:L-L | rs1116907 | No | Yes | Yes | N/A |
| ATP6 | 8655 T>C | syn:I-I | rs2853822 | No | Yes | Yes | N/A |
| ATP6 | 8701 G>A | **non-syn:T-A** | rs2000975 | **No** | **Yes** | **No** | Neuromuscular disorders, somatic lung cancer, thyroid cancer, fertilization failure, atherosclerosis, irritable bowel syndrome, osteosarcoma, MELAS, cardiomyopathy, Parkinson disease, hearing loss |
| ATP6 | 8794 C>T | **non-syn:H-Y** | rs2298007 | **No** | **Yes** | **No** | Homoplasmic variant, coronary arteriosclerosis, irritable bowel syndrome |
| ATP6 | 9042 C>T | syn:H-H | rs3020605 | Yes | No | No | Homoplasmic variant |
| CO3 | 9347 A>G | syn:L-L | rs2853824 | Yes | No | No | Homoplasmic variant |
| CO3 | 9540 C>T | syn:L-L | rs2248727 | No | Yes | No | Normal-Tension Glaucoma (NTG) |
| ND3 | 10398 G>A | **non-syn:T-A** | rs2853826 | **No** | **Yes** | **No** | IVF failure, Fuch's Endothelial Corneal Dystrophy (FECD), breast cancer, esophageal cancer, aging, diabetes, osteoarthritis, schizophrenia, LHON |
| ND4L | 10589 G>A | syn:L-L | rs2853487 | Yes | No | No | Homoplasmic variant |
| ND4L | 10688 A>G | syn:V-V | rs2853488 | No | Yes | Yes | N/A |
| ND4 | 10810 C>T | syn:L-L | rs28358282 | No | Yes | Yes | N/A |
| ND4 | 10873 C>T | syn:P-P | rs2857284 | No | Yes | No | Ischemic stroke |
| ND4 | 12007 G>A | syn:W-W | rs2853497 | Yes | No | No | Homoplasmic variant |
| ND5 | 12705 T>C | syn:I-I | rs193302956 | No | Yes | No | Neuropathy, dystonia, myoclonic epilepsy, MERRF, PD, schizophrenia, bipolar disorder, infantile cardiomyopathy, diabetes, hypertension, AD, Leigh syndrome, blood pressure |
| ND5 | 12720 A>G | syn:M-M | rs2853500 | Yes | No | No | Homoplasmic variant |
| ND5 | 13105 G>A | **non-syn:I-V** | rs2853501 | **No** | **Yes** | **Yes** | Deafness, neuropathy |
| ND5 | 13506 T>C | syn:Y-Y | rs2857287 | No | Yes | Yes | N/A |
| ND5 | 13650 T>C | syn:P-P | rs2854123 | No | Yes | Yes | N/A |
| CYB | 15301 A>G | syn:L-L | rs193302991 | No | Yes | No | Homoplasmic variant |
| CYB | 15301 G>A | syn:L-L | rs193302991 | No | Yes | Yes | Homoplasmic variant |
| D-Loop | 16129 A>G | Noncoding | rs41534744 | No | Yes | Yes | Ewing's sarcoma, predisposition to breast cancer, nasopharyngeal carcinoma |

**Table 2** (*continued*)

| Loci: MT- | SNP | AA Change | rs# | Present in L Cybrids | Present in A+B Cybrids | Present in D Cybrids | Related pathogenesis |
|---|---|---|---|---|---|---|---|
| D-Loop | 16187 T>C | Noncoding | ND | No | Yes | Yes | Friedreich's ataxia, bladder cancer, lung cancer |
| D-Loop | 16189 T>C | Noncoding | rs28693675 | No | Yes | No | Diabetes mellitus, insulin resistance, cardiomyopathy, hypertension, Friedreich's ataxia, Ewing's sarcoma, obesity, predisposition to breast cancer, nasopharyngeal carcinoma |
| D-Loop | 16189 C>T | Noncoding | rs28693675 | No | Yes | Yes | Diabetes mellitus, insulin resistance, cardiomyopathy, hypertension, Friedreich's ataxia, Ewing's sarcoma, obesity, predisposition to breast cancer, nasopharyngeal carcinoma |
| D-Loop | 16223 T>C | Noncoding | rs2853513 | No | Yes | No | Mitochondrial disease , predisposition to ataxia, bladder cancer, predisposition to breast cancer, lung cancer, nasopharyngeal carcinoma, stomach cancer, liver cancer |
| D-Loop | 16278 T>C | Noncoding | rs41458645 | No | Yes | Yes | Bladder cancer, predisposition to breast cancer, lung cancer |
| D-Loop | 16290 C>T | Noncoding | ND | No | Yes | No | Parkinson's disease |
| D-Loop | 16311 C>T | Noncoding | rs34799580 | No | Yes | Yes | Bladder cancer, predisposition to breast cancer, lung cancer |
| D-Loop | 16319 G>A | Noncoding | rs35105996 | No | Yes | No | Homoplasmic variant |
| D-Loop | 16362 T>C | Noncoding | rs62581341 | No | No | Yes | Homoplasmic variant, mitochondrial disease, bladder cancer , predisposition to breast cancer, lung cancer, stomach cancer, liver cancer |

**Notes.**

ND, Not Determined.

All SNPs had a Quality (A Phred-scaled quality score assigned by the variant caller) Score of 100 and Passed all the Filters.

mm) wavelengths. Intact mitochondria with normal $\Delta\Psi$m appeared red, while cells with decreased $\Delta\Psi$m were in a green fluorescent state ($n = 7$ wells per treatment condition).

The ARPE-19 cells (ATCC, Manassas, VA); (Invitrogen-Gibco, Carlsbad, CA), which are the wildtype of the Rho*0* receipt cells of the cybrids, were cultured in Dulbecco's modified Eagle's and Ham's nutrient mixture F-12 (1:1 mixture, vol/vol), 0.37% sodium bicarbonate, 0.058% L-glutamine, 10% fetal bovine serum, antibiotics (100 U/ml penicillin G, 0.1 mg/ml streptomycin sulfate, 10 μg/mL gentamicin), 10 mM non-essential amino acids, and an anti-fungal agent (amphotericin-B 2.5 μg/mL). ARPE-19 cells are a homogeneous, retinal-derived cell line with functional and structural properties similar to human retinal pigment epithelial (RPE) cells (*Dunn et al., 1996*). All cells within the ARPE-19 culture express RPE-specific markers such as CRALBP, BEST1, and RPE-65 (*Moustafa et al., 2017*).

## Cytotoxicity assay - lactate dehydrogenase assay (LDH assay)

L, [A+B], and D cybrids were plated in 96-well plates (10,000 cells/ well) for 24 h and treated with 0 or 40 µM of cisplatin for another 48 h. 50 µM of supernatant containing the released LDH from each corresponding well was transferred to a new 96-well plate, and 50 µM of reaction mixture was subsequently added and gently mixed. The plates were incubated at room temperature for 30 min, and the resulting reactions were stopped by adding 50 µM of stop solution to each well. Absorbance readings were taken at both 490 nm and 680 nm using an absorbance reader (BioTek, ELX 808, Winooski, VT). Percent cytotoxicity levels were calculated using the equation (Treated LDH level–spontaneous LDH level) divided by (Maximum LDH level—spontaneous LDH level) multiplied by one hundred. Average percent cytotoxicity levels were calculated for each ethnic population, normalized to the L haplogroup, and then compared using a two-tailed $t$-test. $P \leq 0.05$ was deemed statistically significant. Cytotoxicity levels were analyzed, each treatment condition was run in quadruplicate and the entire experiment was repeated three separate times.

## RNA isolation, quantitative-real time PCR (qRT-PCR)

L, [A+B], and D cybrids were plated in six-well plates (500,000 cells/well) and incubated for 24 h. Cybrid cells were then treated with culture media containing either 0 or 40 µM of cisplatin for an additional 48 h. Trypsinized cells were pelleted, and RNA was isolated according to the manufacturer's protocol (RNeasy Kit, Qiagen, Valencia, CA). Following RNA quantification (Nanodrop 1000, Thermoscientific, Wilmington, DE), the cDNA was synthesized from 2 µg of RNA (QuantiTect Reverse Transcription Kit, Qiagen), and used for qRT-PCR (StepOnePlus instrument; Applied Biosystems, Carlsbad, CA). Proprietary, predesigned and validated SYBR Green-based primer assays were used (QuantiTect Primer Assays, Qiagen). Qiagen, the source for the primer assays does not release the sequence of their primers and an internal reference gene number (rs#) is not provided, however, the identity of each primer assay can be identified via the gene's GenBank accession number (NM) (https://geneglobe.qiagen.com/product-groups/quantitect-primer-assays). Cancer-related nuclear genes were selected as they are typically the target for many chemotherapeutic treatments and presently have medications that target them. They were examined to understand how variations in mtDNA haplogroups affect retrograde signaling (mitochondria to nucleus) in cells that only vary in their mtDNA. The genes analyzed included *ALK* (Anaplastic Lymphoma Receptor Tyrosine Kinase, NM_004304), *BRCA1* (DNA Repair Associated, NM_007294), *EGFR* (Epidermal Growth Factor Receptor 1, NM_005228), and *ERBB2/HER2* (Erb-b2 Receptor Tyrosine Kinase 2, NM_004448) genes. Similar to the study done by Patel et al. target cycle thresholds (Ct) values were first compared to the Ct values of reference genes, and then, comparisons between untreated and cisplatin-treated values ($\Delta\Delta$Ct) were evaluated for statistical significance. Fold differences were quantified using the equation $2^{(\Delta\Delta Ct)}$, and the values for each sample were normalized to that of the untreated. This was done separately for each haplogroup in order to identify a starting base expression level and observe the gene expression fold difference after cisplatin treatment (Tables 3–4) (*Patel et al., 2019*).
**Table 3 L, [A+B], and D mtDNA haplogroup cybrids each displayed unique overall results in cell and gene expression studies.** Results from the cell and gene expression studies were congregated into one table. As illustrated, each haplogroup had a unique result when observed in large as a summation of the assays and gene expression results. As demonstrated from the cell and gene expression studies, L cybrids had decreased cellular metabolic activity and ALK gene expression after cisplatin treatment ($p = 0.0117$, $p < 0.0001$ respectively). [A+B] cybrids had decreased cellular metabolic activity, ALK gene expression, and EGFR gene expression after cisplatin treatment ($p = 0.0285$, $p = 0.0001$, $p = 0.0246$, respectively). D cybrids had decreased cellular metabolic activity after cisplatin treatment ($p = 0.0001$). *$p < 0.05$; **$p < 0.01$; ***$p < 0.001$.

| | Haplogroup | | |
| --- | --- | --- | --- |
| | **L** | **[A+B]** | **D** |
| **Untreated vs. Cisplatin Treated** | | | |
| Cell Viability | ↓ * | ↓ * | ↓ *** |
| Mitochondrial Membrane Potential | — | — | — |
| LDH Cytotoxicity | L | [A+B]  (* between L and [A+B]; NS between [A+B] and D; NS between L and D) | D |
| **Gene Expression Untreated to Treated** | | | |
| ALK | ↓ **** | ↓ *** | — |
| BRAC1 | — | — | — |
| EGFR | — | ↓ * | — |
| ERBB2/ HER2 | — | — | — |

## Statistical analysis

Statistical analysis of the data were performed by ANOVA (GraphPad Prism, version 5.0). Newman-Keuls multiple-comparison or the two-tailed $t$-tests were used to compare the data within each experiment. Results were normalized to that of the average untreated-L cybrid for each cell study. $P \leq 0.05$ was considered statistically significant. Error bars representing standard deviation (SD) as well as standard error of the mean (SEM) have been graphed for each assay (*Patel et al., 2019*).

**Table 4   L, [A+B], and D mtDNA haplogroup cybrids had varying gene expression values after treatment with cisplatin.** Gene expression results, as well as detailed overviews for *ALK, BRCA1, EGFR*, and *ERBB2/HER2* genes, were summarized into one table. Gene expression values were observed with qPCR for all L, [A+B], and D cybrids and assessed for statistical significance relative to their untreated controls. Cybrids cultured to the fifth passage were plated in six-well plates (500,000 cells/well) and incubated for 24 h. Cybrids were then treated with culture media containing either 0 or 40 µM of cisplatin for an additional 48 h. RNA was isolated and qRT-PCR was conducted to analyze gene expression. Cancer-related nuclear genes were selected as they are common targets and biomarkers in chemotherapeutic treatments. Fold differences were quantified using the equation $2^{(\Delta\Delta Ct)}$, and the values for each sample were normalized to that of the untreated for each haplogroup. There were significant drops in gene expression levels in the treated-L cybrids for *ALK* gene expression ($-18.8\% \pm 9.7\%$, $p < 0.0001$), and treated-[A+B] for both *ALK* and *EGFR* gene expression levels ($-17.5\% \pm 9.5\%$, $p = 0.0001$; $-49.2\% \pm 17.1\%$, $p = 0.025$). Each experiment was conducted three separate times. *$p < 0.05$; **$p < 0.01$; ***$p < 0.001$ (all in bold).

| SYMBOL | GENE NAME | GENBANK ACCESSION NUMBERS | FUNCTIONS | L Cybrids Untreated# vs. Treated P value, Fold | [A+B] Cybrids Untreated# vs. Treated P value, Fold | D Cybrids Untreated# vs. Treated P value, Fold |
|---|---|---|---|---|---|---|
| **ALK** | Anaplastic Lymphoma Receptor Tyrosine Kinase | NM_004304 | Receptor tyrosine kinase that plays an important role in the development of the brain. Found to be rearranged, mutated, or amplified in a series of tumors resulting in many types of cancers. | **<0.0001****, 0.188 ± 0.0973** | **0.0001***, 0.175 ± 0.0950** | 0.290, 2.82 ± 1.48 |
| **BRCA1** | DNA Repair associated | NM_007294 | Nuclear Phospho-protein that acts as a tumor suppressor by maintaining genomic stability. Involved in transcription, DNA, repair of double-stranded breaks, recombination. | 0.356, 1.36 ± 0.372 | 0.294, 1.55 ± 0.477 | 0.158, 3.42 ± 1.39 |
| **EGFR** | Epidermal Growth Factor Receptor | NM_005228 | Triggers cell proliferation when bound to epidermal growth factor. | 0.301, 0.760 ± 0.217 | **0.0246*, 0.492 ± 0.171** | 0.218, .602 ± 0.273 |
| **ERBB2/HER2** | Erb-b2 Receptor Tyrosine Kinase 2 | NM_004448 | Members of epidermal growth factor receptor family of receptor tyrosine kinases. Stabilizes binding of epidermal growth factors to receptor. | 0.839, 1.06 ± 0.298 | 0.379, 0.820 ± 0.189 | 0.325, 1.96 ± 1.75 |

## RESULTS

### Creation of L, A, B, and D cybrids

Figure 1 represents the cybrid cell creation process resulting in cells with identical nuclei but differing mtDNA haplogroups. The mtDNA haplogroups for each subject and cybrid cell line were verified using polymerase chain reaction (PCR) along with restriction enzyme digestion and mtDNA sequencing as described previously in the study conducted by Patel et al. on H and J mtDNA haplogroup cybrid cells (L cybrids, $n = 7$; [A+B] cybrids, $n = 4$; and D cybrids, $n = 3$) (*Patel et al., 2019*).

### Sequencing of mtDNA from L, A, B, and D cybrids

The entire mtDNA from the L, A, B, and D cybrids were sequenced using Next Generation Sequencing (NGS) technology. Table 1 summarizes the mtDNA haplogroup, age and
gender for the subjects in this study. Figures 2–4 show individual mtDNA SNPs that define the haplogroup classification of each cybrid. The private SNPs are those that do not define the L, A, B, or D haplogroups (non-haplogroup defining). The unique SNPs are those SNPs not listed in http://www.mitomap.org or other programs (Figs. 2–4, Table 1) (*Patel et al., 2019*). Values in the cell and gene expression studies are averages and do not represent any particular SNP variant within one haplogroup. They are instead representative of the haplogroup-defining SNPs common among all cybrids within a haplogroup. SNP variants associated with the L, [A+B], and D mtDNA haplogroups as well as the location of the mtDNA nucleotide change, the base pairs changed, their associated rs number, and any associated pathogenesis found in the literature were identified and assessed. Non-synonymous changes present in some but not all haplogroups tested were found at m.4824 A > G, rs ND; m.5178 C > A, rs28357984; m.5442 T > C, rs3020601; m.7146 G > A, rs372136420; m.8701 G > A, rs2000975; m.8794 C > T , rs2298007; m.10398 G > A, rs2853826; and m.13105 G > A, rs2853501 (http://www.hmtvar.uniba.it and http://www.phylotree.org) (Table 2).

## Relative cellular metabolic activity of untreated L, A+B, and D cybrids

The cellular metabolic activity measured with the MTT assay is representative of the cell viability within the cybrid cultures. The untreated cybrids, representing different maternal ethnic populations (African (L), Hispanic [A+B], or Asian (D) mtDNA haplogroups), had statistically similar relative cellular metabolic activities. All values were normalized to the average of the untreated-L cybrids and displayed as the mean $\pm$ SEM (Fig. 5A). The distribution of each value was also presented in a dot-plot graph representing mean $\pm$ SD (Fig. 5B). The untreated-L cybrids (102.4% $\pm$ 7.0% SEM; SD = 21.0%, $n = 7$) had similar relative cellular metabolic activity when compared to the untreated-D cybrids (111.7% $\pm 4.3\%$ SEM; SD = 11.3%, $n = 3$, $p = 0.311$) and the untreated-[A+B] cybrids (95.0% $\pm$ 8.5% SEM; SD = 20.9%, $n = 4$, $p = 0.512$). Untreated-[A+B] cybrids showed similar cellular metabolic activity compared to the untreated-D cybrids ($p = 0.0933$). Experiments were analyzed in quadruplicate, and the entire experiment was repeated twice.

## Relative cellular metabolic activity of L, A+B, and D cybrids after cisplatin treatment

We found significantly reduced cellular metabolic activity after cisplatin treatment in the L, [A+B], and D cybrids compared to their respective untreated controls (Figs. 6A 6B). All values were normalized to the average of the untreated-L cybrids and displayed as the mean $\pm$ SEM (Fig. 6A). The distribution of each value was also presented in a dot-plot graph representing mean $\pm$ SD (Fig. 6B). The cisplatin-treated-L cybrids showed the greatest reduction in relative cellular metabolic activity compared to the untreated-L cybrids (75.0% $\pm$ 6.7% SEM; SD = 20.0%, $n = 7$; 102.4% $\pm 7.0\%$ SEM; SD = 21.0%, $n = 7$, respectively) ($p = 0.0117$). The cisplatin-treated-D cybrids showed a significant decrease in cell viability compared to the untreated-D cybrids (86.9% $\pm$ 1.6% SEM; SD = 4.3%, $n = 3$; 111.7% $\pm$ 4.3% SEM; SD = 11.3%, $n = 3$, respectively) ($p = 0.0001$). The cisplatin-treated-[A+B] cybrids also had a statistically significant decrease in cellular metabolic activity compared

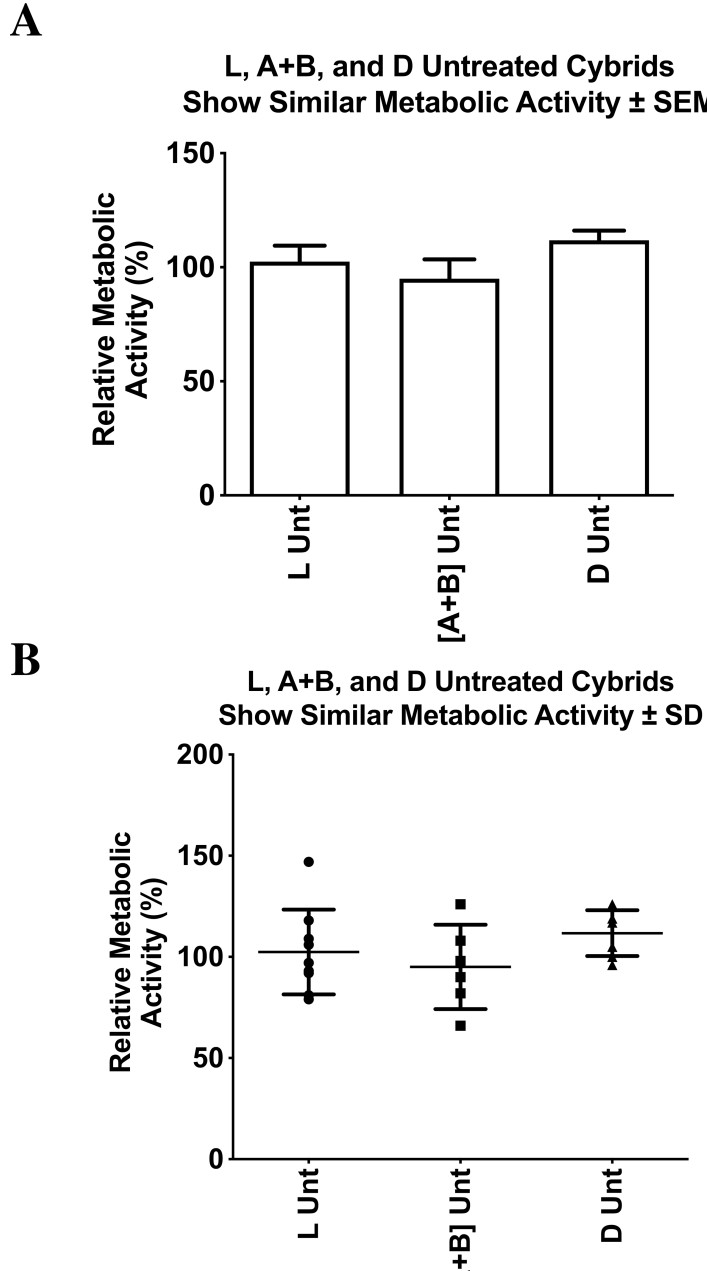

**A**

**L, A+B, and D Untreated Cybrids
Show Similar Metabolic Activity ± SEM**

**B**

**L, A+B, and D Untreated Cybrids
Show Similar Metabolic Activity ± SD**

**Figure 5** **L, [A+B], and D untreated cybrids had similar cellular relative metabolic activity.** Cybrid cell lines of the fifth passage, containing mitochondria with either L, [A+B], or D haplogroups, were cultured in flasks until confluent. They were then plated on a 96-well plate with each well containing 50,000 cells and incubated at 37 °C for another 48 h. The MTT assay was then conducted to assess relative cellular metabolic activity. Absorbance was measured on an absorbance reader at 570 nm (measured) and 630 nm (reference). Cybrids had statistically similar relative cellular metabolic activities. (A) All values were normalized to the average of the untreated-L cybrids and displayed as the mean ± Standard Error Margin (SEM). (continued on next page...)

**Figure 5 (…continued)**
(B) The distribution of each value was also presented in a dot-plot graph representing mean $\pm$ Standard Deviation (SD). The untreated-L cybrids (102.4% $\pm$ 7.0% SEM; SD = 21.0%, $n = 7$) had similar relative cellular metabolic activity when compared to the untreated-D cybrids (111.7% $\pm$ 4.3% SEM; SD = 11.3%, $n = 3$, $p = 0.311$) and the untreated-[A+B] cybrids (95.0% $\pm$ 8.5% SEM; SD = 20.9%, $n = 4$, $p = 0.512$). Untreated-[A+B] cybrids showed similar cellular metabolic activity compared to the untreated-D cybrids ($p = 0.0933$). Experiments were analyzed in quadruplicate, and the entire experiment was repeated twice. $*p < 0.05$; $**p < 0.01$; $***p < 0.001$.

to their respective untreated-[A+B] cybrids (70.3% $\pm$ 4.5% SEM; SD = 11.1%, $n = 4$; 95.0% $\pm$ 8.5% SEM; SD = 20.9%, $n = 4$, respectively) ($p = 0.0285$) (Figs. 6A and 6B). Each treatment was run in quadruplicate and the entire experiment was repeated twice.

## Mitochondrial membrane potential ($\Delta\Psi$m) after cisplatin treatment

Mitochondrial membrane potentials for untreated and treated cybrid cells were measured. All values were normalized to the average of the untreated-L cybrids and displayed as the mean $\pm$ SEM (Fig. 7A). The distribution of each value was also presented in a dot-plot graph representing mean $\pm$ SD (Fig. 7B). Results indicated that there was no significant difference in mitochondrial membrane potential in the cisplatin-treated cybrids compared to the untreated cybrids. Cisplatin-treated-L cybrids had a relative mitochondrial membrane potential of 108.4% $\pm$ 8.2% SEM; SD = 21.8% versus untreated-L cybrids (101.0% $\pm$ 11.2% SEM; SD = 29.5%, $n = 7$, $p = 0.602$). Cisplatin-treated-[A+B] displayed 136.5% $\pm$ 33.9% SEM; SD = 67.8% versus untreated-[A+B] cybrids (147.5% $\pm$ 42.0% SEM; SD = 84.0%, $n = 4$, $p = 0.845$). The cisplatin-treated-D cybrids were observed to have a relative mitochondrial membrane potential of 115.0% $\pm$ 3.5% SEM, SD = 7.0% versus the untreated-D cybrids (127.3% $\pm$ 20.1% SEM; SD = 40.3%, $n = 3$, $p = 0.571$) (Figs. 7A and 7B). Each treatment was run in quadruplicate and the entire experiment was repeated twice.

## Mitochondrial membrane potential ($\Delta\Psi$m) assay (IncuCyte® live cell analyzer with ARPE-19 cells)

After cisplatin treatment, the cybrids showed decreased metabolic activity but the mitochondrial membrane potentials remained unchanged, which was unexpected. Therefore, we conducted additional experiments using the ARPE-19 cell line that allowed for comparison of growth/proliferation over 40 h in cultures treated with 20 $\mu$M or 40 $\mu$M cisplatin ($n = 7$ per treatment condition). All values were displayed as the mean $\pm$ SD. Over the first 8 h of culture, the untreated and cisplatin-treated ARPE-19 cells showed similar rates of growth (Fig. 8B). By 10 h the untreated ARPE-19 cells showed an increasing slope of growth compared to the 20 $\mu$M and 40 $\mu$M cultures that were stagnant in their growth. By 40 h, the untreated had a 2-fold higher number of cells than cisplatin-treated cultures. (Untreated 1969 $\pm$ 115 standard deviation (SD)), versus 20 $\mu$M (1023 $\pm$ 81 SD, $p = 0.0006$) and 40 $\mu$M (945 $\pm$ 74 SD, $p = 0.0001$) (At 40 h, the $\Delta\Psi$m levels were similar between the untreated AREP-19 (1.8993 $\pm$ 0.145 SD) versus 20 $\mu$M (2.0698 $\pm$ 0.161 SD, $p =$ NS) and 40 $\mu$M (2.0520 $\pm$ 0.162 SD, p = NS) cisplatin-treated ARPE-19 cultures (Fig. 8A).

**A**

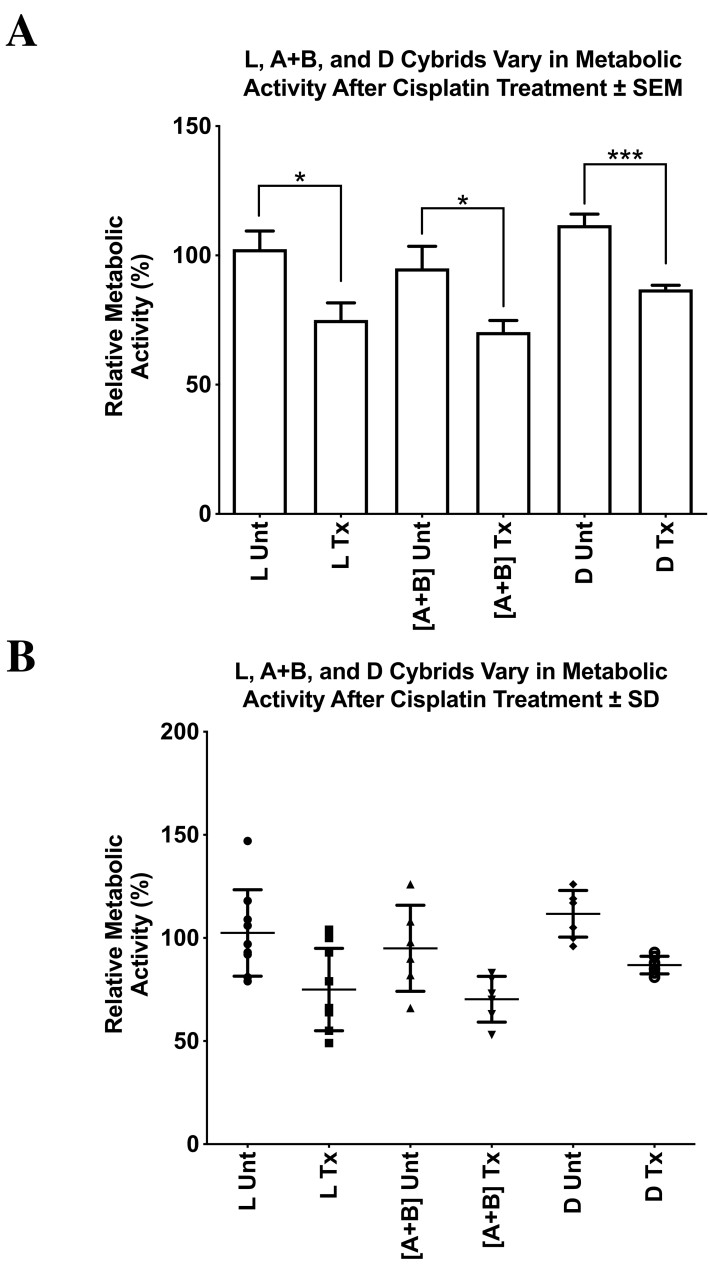

**Figure 6  L, [A+B], and D cybrids vary in relative cellular metabolic activity after cisplatin treatment.**
L, [A+B], and D cybrids in the fifth passage were cultured and then plated on 96-well plates with each well containing 50,000 cells. After 24 h, media were removed, and cells were treated with 100 µL of 40 µM concentration of cisplatin. Plates were then incubated at 37 °C for another 48 h. The MTT assay was conducted to assess relative cellular metabolic activity. Absorbance values were measured on an absorbance reader at 570 nm (measured) and 630 nm (reference). (A) All values were normalized to the average of the untreated-L cybrids and displayed as the mean ± Standard Error Margin (SEM). The distribution of each value was also presented in a dot-plot graph representing mean ± Standard Deviation (SD). The cisplatin-treated-L cybrids showed the greatest reduction in relative cellular metabolic activity compared to the untreated-L cybrids 

**Figure 6 (...continued)**
(B) (75.0% ± 6.7% SEM; SD = 20.0%, $n = 7$; 102.4% ± 7.0% SEM; SD = 21.0%, $n = 7$, respectively) ($p = 0.0117$). The cisplatin-treated-D cybrids showed a significant decrease in cell viability compared to the untreated-D cybrids (86.9% ± 1.6% SEM; SD = 4.3%, $n = 3$; 111.7% ± 4.3% SEM; SD = 11.3%, $n = 3$, respectively) ($p = 0.0001$). The cisplatin-treated-[A+B] cybrids also had a statistically significant decrease in cellular metabolic activity compared to their respective untreated-[A+B] cybrids (70.3% ± 4.5% SEM; SD = 11.1%, $n = 4$; 95.0% ± 8.5% SEM; SD = 20.9%, $n = 4$, respectively) ($p = 0.0285$). Each treatment was run in quadruplicate and the entire experiment was repeated twice. *$p < 0.05$; **$p < 0.01$; ***$p < 0.001$.

## Lactate dehydrogenase cytotoxicity after cisplatin treatment

The cytotoxicity levels of each of the cybrids were measured with the lactate dehydrogenase (LDH) assay. All values were normalized to the average of the untreated-L cybrids and displayed as the mean ± SEM (Fig. 9A). The distribution of each value was also presented in a dot-plot graph representing mean ± SD (Fig. 9B). When normalized to the L cybrids, there were significantly higher LDH cytotoxicity levels in L cybrids compared to [A+B] cybrids (111.5% ± 16.5% SEM; SD = 40.5%, $n = 7$ versus 41.8% ± 9.8% SEM; SD = 16.9%, $n = 4$, respectively) ($p = 0.0270$). The D cybrids trended lower LDH levels but they were not significant (48.9% ± 20.2% SEM; SD = 35.0% , $n = 3$, $p = 0.0576$) compared to the L cybrids. LDH cytotoxicity levels between the [A+B] and D cybrids were not statistically different ($p = 0.766$) (Figs. 9A and 9B). The entire experiment was repeated three separate times.

## Gene expression levels in L, [A+B], and D cybrids after cisplatin treatment

*ALK* (Anaplastic Lymphoma Kinase): Transcription levels for L cybrids and the combined [A+B] cybrids dropped significantly after cisplatin treatment (0.18 fold ± 0.097, $n = 7$, $p < 0.0001$; 0.17 fold ± 0.095, $n = 4$, $p = 0.0001$, respectively), while the D cybrids had no significant change in *ALK* gene expression ($n = 3$, $p = 0.290$) (Tables 3–4).

*BRCA1* (Breast Cancer 1): Gene expression levels for all cybrids (L, [A+B], and D) did not significantly change after treatment with cisplatin compared to untreated-control cybrids ($n = 7$, $n = 4$, $n = 3$, $p = 0.356$, $p = 0.294$, $p = 0.158$, respectively) (Tables 3–4).

*EGFR* (Epidermal Growth Factor Receptor 1): After cisplatin treatment, cybrids containing L and D mtDNA haplogroups had no significant change in *EGFR* gene expression compared to their respective untreated-controls ($n = 7$, $p = 0.301$ and $n = 3$, $p = 0.218$, respectively). However, cisplatin-treated cybrids containing [A+B] mtDNA haplogroups significant decreased in *EGFR* gene expression levels when compared to untreated-[A+B] cybrids (0.50 fold ±0.17, $n = 4$, $p = 0.0246$) (Tables 3–4).

*ERBB2/HER2* (Erb-b2 Receptor Tyrosine Kinase 2): Similar to the *BRCA1* gene expression levels, *ERBB2/HER2* gene expression levels were not significantly different after cisplatin treatment in any of the cybrid cell lines (L, [A+B], and D) tested ($n = 7$, $n = 4$, $n = 3$; $p = 0.839$, $p = 0.379$, $p = 0.325$, respectively) (Tables 3–4).

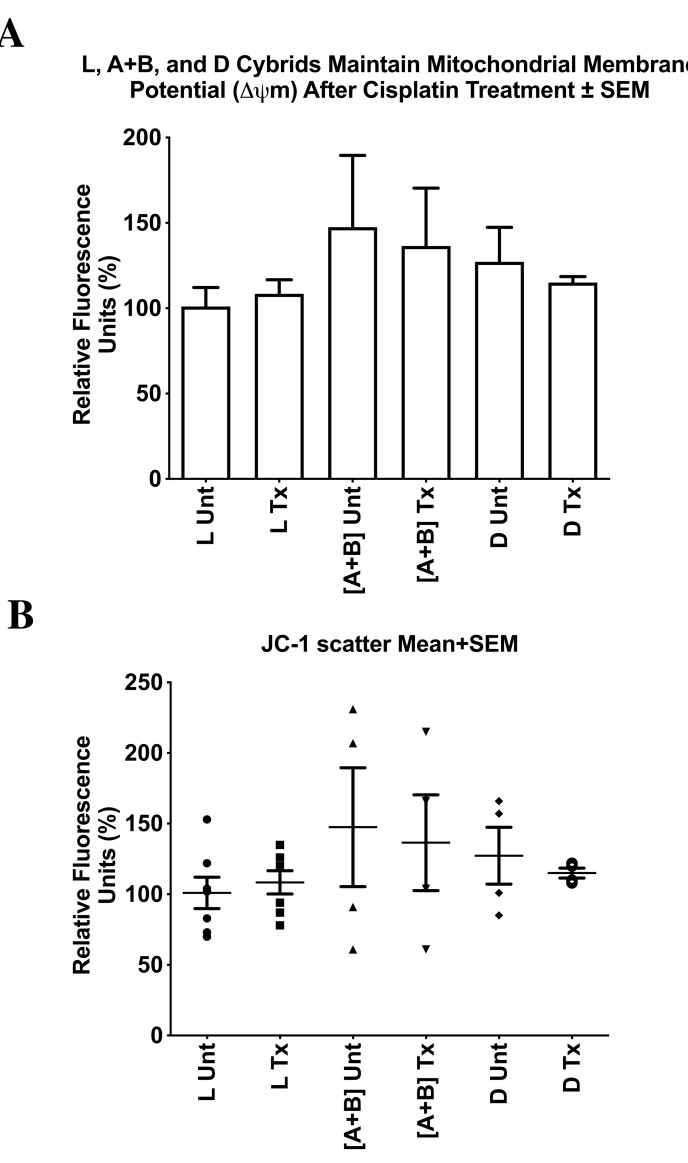

**A**

**L, A+B, and D Cybrids Maintain Mitochondrial Membrane Potential (Δψm) After Cisplatin Treatment ± SEM**

**B**

**JC-1 scatter Mean+SEM**

**Figure 7** **L, [A+B], and D cybrids maintained mitochondrial membrane potential (ΔΨm) after cisplatin treatment.** Mitochondrial Membrane Potentials for untreated and treated cybrid cells were measured. Mitochondrial Membrane Potentials for untreated and treated cybrid cells were measured. L, [A+B], and D cybrids were cultured to the fifth passage and then plated in 24-well plates (100,000 cells/well), incubated 24 h and treated with 0 or 40 μM of cisplatin for another 48 h. The JC-1 assay was conducted and fluorescence values were obtained using a microplate reader. All values were normalized to the average of the untreated-L cybrids. (A) All values were normalized to the average of the untreated-L cybrids and displayed as the mean ± Standard Error Margin (SEM). (B) The distribution of each value was also presented in a dot-plot graph representing mean ± Standard Deviation (SD). Results indicated that there was no significant difference in mitochondrial membrane potential after cisplatin treatment in L, [A+B], or D cybrids. Cisplatin-treated-L cybrids had a relative mitochondrial membrane potential of 108.4% ± 8.2% SEM; SD = 21.8% versus untreated-L cybrids (101.0% ± 11.2% SEM; SD = 29.5%, $n = 7$, $p = 0.602$). Cisplatin-treated-[A+B] displayed 136.5% ± 33.9% SEM; (continued on next page…)

**Figure 7 (…continued)**
SD = 67.8% versus untreated-[A+B] cybrids (147.5% ± 42.0% SEM; SD = 84.0%, $n = 4$, $p = 0.845$). The cisplatin-treated-D cybrids were observed to have a relative mitochondrial membrane potential of 115.0% ± 3.5% SEM; SD = 7.0% versus the untreated-D cybrids (127.3% ± 20.1% SEM; SD = 40.3%, $n = 4$, $p = 0.571$) (A). Each treatment was run in quadruplicate and the entire experiment was repeated twice. $p < 0.05$; $**p < 0.01$; $***p < 0.001$.

## DISCUSSION

### Sequencing of the entire mtDNA

The NGS technology used to analyze the mtDNA from each of the L, A, B, and D cybrids showed that the majority of SNPs identified were haplogroup defining. The private SNPs (non-haplogroup defining), unique SNPs (not listed in http://www.mitomap.org), and heteroplasmy SNPs were found in individual cybrids and not throughout all of the L, A, B, or D cybrids. Similar to the study comparing cybrids with H and J mtDNA haplogroups, this suggests that the differential retrograde signaling between the L, A, B, and D mtDNA haplogroups is due to the accumulation of the haplogroup defining SNPs rather than a single mutation or private SNP (*Patel et al., 2019*). These data are also consistent with another cybrid study which used allelic discrimination and Sanger sequencing to identify the mtDNA haplogroups (*Atilano et al., 2015*). NGS technology was used for deep sequencing of the mtDNA (ranging from 1,000 to 100,000, with an average depth of 30,000), which allowed for low-level heteroplasmy to be reliably identified (*Patel et al., 2019*). This method helps distinguish artifact from low-level heteroplasmy because both strands of mtDNA are independently sequenced in both directions. The role of mtDNA haplogroups in retrograde signaling is still under investigation, and many pathways may still be unidentified.

Haplogroup-defining SNPs were analyzed and the nucleotide changes between the L, [A+B], and D cybrids were identified (Table 2). Non-synonymous changes (causing a change in amino acid) present in some but not all haplogroups tested were found at eight locations. Related pathogenesis to these SNP variants have been associated with diseases such as asthenospermia, atherosclerosis, breast cancer, cardiomyopathy, esophageal cancer, fertilization failure, Fuch's endothelial corneal dystrophy (FECD), glaucoma, diabetes, hypertension, infertility, irritable bowel syndrome, myocardial infarction, neuropathy, osteoarthritis, osteosarcoma, Parkinson's, mobility impairment, neuromuscular disorders, schizophrenia, somatic lung cancer, and thyroid cancer (http://www.hmtvar.uniba.it and http://www.phylotree.org) (Table 2). One could speculate that these SNP variants may play a role in the differential susceptibility to cisplatin in L, [A+B], and D cybrids because the cybrids have identical nuclei but vary in their mtDNA. Additional studies are needed to better understand the relationship between mtDNA haplogroups and drug susceptibility-related cellular pathways.

**A**

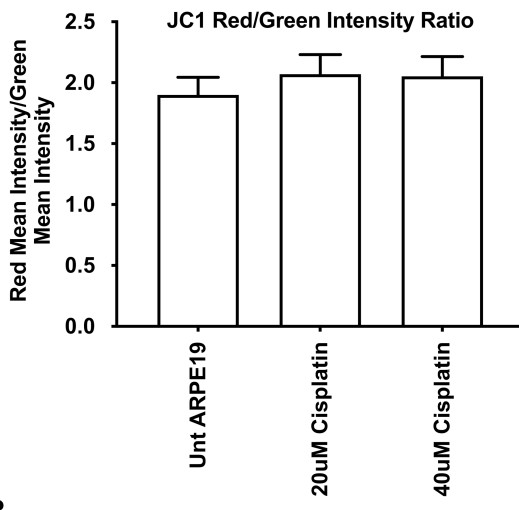

**B**

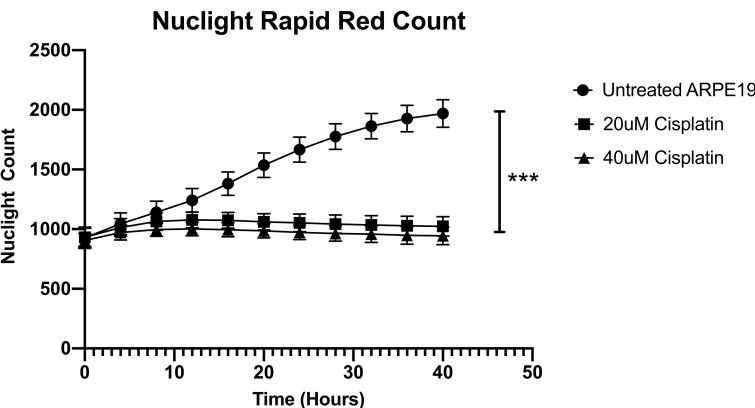

**Figure 8** **ARPE-19 Cells maintained mitochondrial membrane potential ($\Delta\Psi$ m) after cisplatin treatment (Incucyte® Live Cell Analyzer).** After cisplatin treatment, the cybrids showed decreased metabolic activity but the mitochondrial membrane potentials remained unchanged, which was unexpected. Therefore, we conducted additional experiments using the ARPE-19 cell line that allowed for comparison of growth/proliferation over 40 h in cultures treated with 20 $\mu$M or 40 $\mu$M cisplatin ($n = 7$ per treatment condition). The mitochondrial membrane potential was measured with the JC-1 reagent that was added to cultures for 15 min. Fluorescence was measured for red and green wavelengths using a microplate reader. (A) All values were displayed as the mean $\pm$ SD. Over the first 8 h of culture, the untreated and cisplatin-treated ARPE-19 cells showed similar rates of growth. (B) By 10 h the untreated ARPE-19 cells showed an increasing slope of growth compared to the 20 $\mu$M and 40 $\mu$M cultures that were stagnant in their growth. By 40 hours, the untreated had a 2-fold higher number of cells than cisplatin-treated cultures. (Untreated 1969 $\pm$ 115 standard deviation (SD)), versus 20 $\mu$M (1023 $\pm$ 81 SD, $p =$ 0.0006) and 40 $\mu$M (945 $\pm$ 74 SD, $p =$ 0.0001) (At 40 h, the $\Delta\Psi$ m levels were similar between the untreated AREP-19 (1.8993 $\pm$ 0.145 SD) versus 20 $\mu$M (2.0698 $\pm$ 0.161 SD, $p =$ NS) and 40 $\mu$M (2.0520 $\pm$ 0.162 SD, $p =$ NS) cisplatin-treated ARPE-19 cultures. $p < 0.05$; **$p < 0.01$; ***$p < 0.001$.

**A**

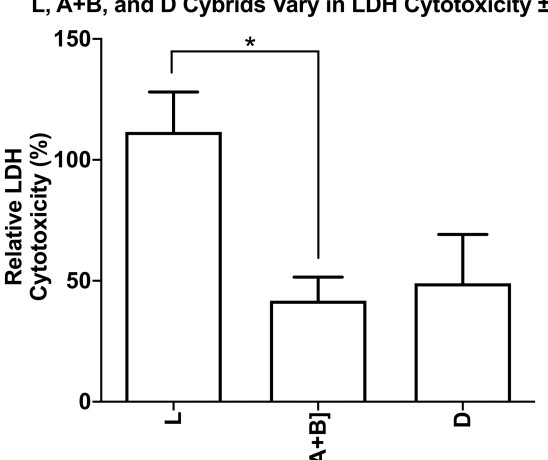

**B**

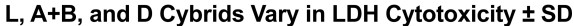

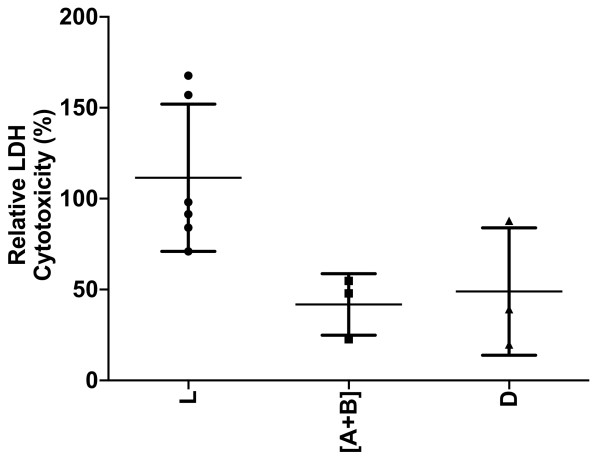

**Figure 9** **L, [A+B], and D cybrids vary in LDH cytotoxicity.** The cytotoxicity levels of L, [A+B], and D cybrids were measured with the lactate dehydrogenase (LDH) assay. Cybrids cultured to the fifth passage were plated in 96-well plates (10,000 cells/ well) for 24 h and treated with 0 or 40μM of cisplatin for another 48 h. The LDH assay was conducted and absorbance values were taken at both 490 nm and 680 nm using an absorbance reader. Percent cytotoxicity levels were calculated using the equation (Treated LDH level −spontaneous LDH level) divided by (Maximum LDH level −spontaneous LDH level) multiplied by one hundred. (A) All values were normalized to the average of the untreated-L cybrids and displayed as the mean ± Standard Error Margin (SEM). (B) The distribution of each value was also presented in a dot-plot graph representing mean ± Standard Deviation (SD). When normalized to the L cybrids, there were significantly higher LDH cytotoxicity levels in L cybrids compared to [A+B] cybrids (111.5% ± 16.5% SEM; SD = 40.5%, $n = 6$ versus 41.8% ± 9.8% SEM; SD = 16.9%, $n = 3$, respectively) ($p = 0.0270$). The D cybrids trended lower LDH levels but they were not significant (48.9% ± 20.2% SEM; SD = 35.0%, $n = 3$, $p = 0.0576$) compared to the L cybrids. LDH cytotoxicity levels between the [A+B] and D cybrids were not statistically different ($p = 0.766$). The entire experiment was repeated three separate times. *$p < 0.05$; **$p < 0.01$; ***$p < 0.001$.

## Variations within mtDNA in L, A+B and D cybrids are associated with a differential decline in relative cellular metabolic activity after treatment with cisplatin

After 48 h, the untreated L (African maternal descent), [A+B] (Hispanic maternal descent), and D (Asian mitochondrial descent) cybrids had similar levels of cellular metabolic activities.

However, in response to cisplatin treatment, there was a differential decline in relative cellular metabolic activity (L >D >[A+B] cybrids) compared to their respective untreated cultures with the [A+B] cybrids showing the smallest decrease in metabolic activity. One can speculate that if a person with mtDNA containing a more resistant SNP variant, such as haplogroup [A+B], was treated with identical doses of cisplatin compared to an individual with haplogroup L mitochondria, the first individual might experience signs of resistance to the drug, while the latter would experience a significant drop in cellular metabolic activity in the targeted cells. These data suggest that an individual's mtDNA haplogroup profile may play a role in their response to cisplatin treatment, thereby affecting its efficiency and severity when used in chemotherapy. Understanding how pharmacogenomics play a role in patient treatment, particularly, how the patient's mtDNA SNP profile is associated with variations in drug susceptibility, would allow future treatment programs to focus more on an individual's genetic background in order to achieve more effective results (*Van Gisbergen et al., 2015*). Further studies will be required to better understand the relationship between mtDNA SNP variants and differences in drug susceptibility.

## Cisplatin treatment induced apoptosis without significantly disrupting mitochondrial membrane potential

Data from the JC-1 assay indicated that when treated with cisplatin, mitochondrial membrane potential in all cybrids was maintained. Surprisingly, L, [A+B], and D cybrids did not show any significant change in mitochondrial membrane potential after cisplatin treatment. OXPHOS related ATP production is dependent on mitochondrial membrane potential, and a drastic decrease or increase in membrane potential in the intermembrane space results in less efficient ATP production via ATP synthase. Cisplatin is known to disrupt mitochondrial membrane potential by inducing the synthesis of pro-apoptotic proteins such as Bcl-2 family proteins, Bax and Bak, which form porous defects within the outer membrane of the mitochondria and result in the release of apoptotic factors such as Cytochrome *c* into the cytosol, ultimately resulting in cell death (*Pabla et al., 2008*). However, a failure of the mitochondrial membrane potential to decrease in the case of cybrids containing L, [A+B], or D mtDNA haplogroups, suggests that a decrease in cell viability leading to apoptosis was induced by a mechanism that did not disrupt the proton gradient within the intermembrane space of the mitochondria. In a previous study, it was observed that somatic mtDNA mutations affect apoptosis without affecting ROS levels or OXPHOS (*Trifunovic et al., 2005*). Additionally, a study on renal cells demonstrated that neither mitochondrial membrane potential nor OXPHOS related ATP production significantly changed at any time during cisplatin exposure (*Cummings & Schnellmann, 2002*).

The ARPE-19 experiments were conducted to verify the cybrid results obtained from the JC-1 mitochondrial membrane potential assays. The ARPE-19 experiments showed the 20 µM and 40 µM cisplatin prevented cell proliferation, indicating growth inhibition and decreased cell health but did not cause elimination of the cells. Surprisingly the levels for mitochondrial membrane potential ($\Delta\Psi$m) were similar in the untreated and cisplatin-treated cultures. This assay relies on the ratio of red (healthy) and green (dead cells) to indicate change in $\Delta\Psi$m. The red fluorescence is proportional to the potential but diminished in apoptotic or necrotic cells. In ARPE-19 cultures, the cisplatin caused cells to stain more brightly with both red and green intensity, so the final ratio was unchanged (Fig. 8A). This suggests that the cisplatin is not lowering the mitochondrial membrane potential, which agrees with Kleih and colleagues who observed that ovarian cancer cells treated with cisplatin also maintained their mitochondrial membrane potential (*Cocetta, Ragazzi & Montopoli, 2019*; *Kleih et al., 2019*). Similarly, English et al. found that cisplatin-treated neuronal cells could maintain and restore their mitochondrial membrane potential through the uptake of cisplatin-resistant mitochondria from astrocytes (*English et al., 2020*).

An alternative pathway leading to apoptosis as a result of cisplatin treatment may be induced by endoplasmic reticulum (ER) stress. A previous study conducted by Gisbergen et al. demonstrated that ER stress-related apoptosis was induced as a result of cisplatin treatment (*Van Gisbergen et al., 2015*). In that study, cisplatin treatment caused ER stress, which activated Caspase-12 cleavage and consequently resulted in apoptosis. Additionally, in a study conducted by Cumming et al. another ER-associated protein leading to apoptosis was identified. Inhibition of $Ca^{2+}$-independent phospholipase A2 (ER-iPLA2) resulted in an increase of cisplatin-induced apoptosis in primary rabbit proximal tubular cultures (*Cummings, McHowat & Schnellmann, 2004*). In a previous study conducted on ARPE cybrid cells containing either H or J mtDNA haplogroups, J cybrids showed a significant decline in mitochondrial membrane potential after cisplatin treatment, while H cybrids did not (*Patel et al., 2019*). A decrease in mitochondrial membrane potential would suggest a decrease in OXPHOS related ATP production and an increase in glycolytic ATP production. As indicated by *Stewart et al. (2011)* $\rho^0$ cells depleted of mitochondria via Ethidium Bromide (EtBr), rely completely on glycolysis for energy production, indicating that OXPHOS inhibition is inversely related to glycolytic ATP production. From the study conducted on H and J cybrids, along with the current study, one can speculate that the unique composition of an individual's mtDNA plays a role in their susceptibility to cisplatin treatment. Additional studies are needed to better understand the relationship between mtDNA variants and drug susceptibility.

## Cybrids with dissimilar mtDNA haplogroups vary in LDH cytotoxicity after cisplatin treatment

Lactate Dehydrogenase cytotoxicity levels measured by the LDH assay illustrated varying results for each of the groups examined. The L cybrids exhibited the highest normalized LDH cytotoxicity levels, which were statistically greater than the [A+B] but not the D cybrids ($p = 0.0270$ and $p = 0.0576$, respectively). Additionally, statistical analysis

between the [A+B] cybrids and the D demonstrated that the two groups did not have statistically different LDH cytotoxicity levels ($p = 0.766$) (Figs. 9A and 9B). Cisplatin-induced cytotoxicity is primarily by the mediation of covalent purine adducts within the nuclear (n) DNA that result in disruption of transcription and replication, consequently leading to cell death (*Dasari & Tchounwou, 2014*). One can speculate that nDNA adduct formation leading to extracellular cytotoxicity is less prevalent in cybrids containing [A+B] mtDNA haplogroups compared to cybrids containing L mtDNA haplogroups. In a previous study conducted by Kohno et al. on cancer cells, it was determined that the prevalence of guanine base sequences within the mtDNA was a determining factor for the ability of cisplatin to bind effectively and form adducts within the DNA, therefore supporting the idea that cisplatin-induced cytotoxicity depended on the unique profile of the individual's mtDNA (*Kohno et al., 2015*). Variations in cisplatin-induced cytotoxicity between cybrids with different mtDNA may suggest that mtDNA SNP variants play a key role in the extent to which adducts form. Additional studies are needed to better understand the relationship between mtDNA haplogroups and DNA adduct formation.

### Variations within mtDNA are associated with modulation of retrograde signaling, resulting in varying nuclear gene expression levels

***Varied gene expression levels among experimented mtDNA haplogroups***

*ALK*: We examined the *ALK* gene as it is a common biomarker in patients with non-small-cell lung cancer (NSCLC) and is the primary target of many chemotherapeutic treatments used to treat NSCLC and anaplastic large cell lymphomas (ALCL). (*Holla et al., 2017*). Cybrids containing L or [A+B] mtDNA haplogroups had significantly decreased ALK gene expression levels compared to their respective untreated controls, while the cybrids containing D mtDNA haplogroups did not have a significant change compared to their respective controls (Tables 3–4). The *ALK* gene codes for a receptor tyrosine kinase in the transmembrane region, and plays an important role in early brain and nervous system development (*Hallberg & Palmer, 2016*; *Webb et al., 2009*). High ALK gene expression values have been associated with neuroblastoma formation, and the presence of non-small cell lung cancers (NSCLC) (*Holla et al., 2017*). ALK inhibition has also been a chemotherapeutic target in treatments of these cancers. In the cell culture studies, the D cybrids had higher trending relative cellular metabolic activity than that of cybrids containing L or [A+B] mtDNA haplogroups. Accelerated cellular metabolic activity is a fundamental characteristic of cancer cells, and such behavior exhibited by cybrids containing D mtDNA haplogroups may represent cancer cell-like traits (*Mizutani et al., 2009*). These data are significant as it suggests that gene expression values in nuclear genes can be altered depending on the particular mtDNA haplogroup present (*Cocetta, Ragazzi & Montopoli, 2019*; *Da Cunha, Torelli & Kowaltowski, 2015*; *Patel et al., 2019*). Our findings support this because all cybrids contained identical nDNA and media culture conditions but differed in their specific mtDNA haplogroups, therefore, it is unlikely that nDNA contributed to the differences in cisplatin susceptibility. This suggests that the variation in mtDNA can modulate retrograde signaling (mitochondria to nucleus) that results in

varied gene expression for the *ALK* gene. The mechanism for this occurrence is not fully understood.

*EGFR:* The *EGFR* gene as it is a biomarker used to evaluate the efficacy of chemotherapeutic treatments for patients with NSCLC (*Wang et al., 2018*). After cisplatin treatment, cybrid cells containing L and D mtDNA haplogroups had no significant change in *EGFR* gene expression compared to untreated cybrids. However, cybrids containing the [A+B] mtDNA haplogroups had a significant change in *EGFR* gene expression levels and dropped 50.8% ± 17.1% when compared to their respective untreated cybrids ($p = 0.0246$) (Tables 3–4). The *EGFR* gene codes for a transmembrane glycoprotein that binds to epidermal growth factors, and expression of this gene is associated with cell proliferation. In an in vivo study on patients with NSCLC, degradation of the *EGFR* gene occurred as a result of platinum-based chemotherapy treatment (*Wang et al., 2018*). Additionally, *Wang et al. (2018)* concluded that the *EGFR* gene became highly unstable as a result of the chemotherapy treatment, and that *EGFR* expression levels dropped significantly in some patients while remaining similar in others. In a different study conducted in vivo, cisplatin treatment resulted in *EGFR* phosphorylation and subsequent ubiquitination and degradation (*Ahsan et al., 2010*). This indicated that those individuals with decreased *EGFR* levels were more susceptible to negative mutations and, as a result, more susceptible to the treatment than those with no significant decrease in *EGFR* levels. Results from the current study suggest that patients with [A+B] mtDNA haplogroups may express certain factors that result in their *EGFR* gene to become significantly more unstable than patients with L or D mtDNA haplogroups after treatment with cisplatin. These data suggest that expression levels of nuclear genes can be altered depending on the particular mtDNA haplogroup present (*Cocetta, Ragazzi & Montopoli, 2019*; *Da Cunha, Torelli & Kowaltowski, 2015*; *Patel et al., 2019*). Our findings show that different mtDNA variants can alter the expression levels the *EGFR* gene.

### Similar gene expression levels among experimented mtDNA haplogroups

*BRCA1:* The *BRCA1* gene was analyzed because it is a common biomarker used to understand the efficacy of chemotherapeutic treatments in patients with breast and ovarian cancers (*Quinn et al., 2007*; *Welcsh & King, 2001*). Gene expression levels for all cybrids (L, [A+B], and D) tested did not significantly change, relative to their respective untreated group, after treatment with cisplatin ($p = 0.356$, 0.294, and 0.158 respectively) (Tables 3–4). The *BRCA1* gene encodes a nuclear phosphoprotein that plays a role in maintaining genomic stability, and germline mutations within this gene have been highly associated with breast and ovarian cancers (*Welcsh & King, 2001*). In an in vivo study conducted by *Quinn et al. (2007)* *BRCA1* gene expression levels were found to be directly correlated with overall survival in patients with ovarian cancer. Results from the current study indicate that the variation in mtDNA within the varying cybrids did not significantly change the *BRCA1* levels of the cybrids. This suggests that the L, [A+B], and D mtDNA haplogroups in ARPE cybrids may not have a significant effect on the gene expression pathways associated with the *BRCA1* gene.

*ERBB2/HER2:* We examined the *ERRB2/HER2* gene as it is a common biomarker used to assess patients with breast cancer (*Birnbaum, Sircoulomb & Imbert, 2009*). Similar

to the *BRCA1* gene expression levels, *ERBB2/HER2* gene expression levels were not significantly different after cisplatin treatment in all of the cybrid cells (L, [A+B], and D) ($p = 0.839, 0.379, 0.325$ respectively) (Tables 3–4). Unlike the *BRCA1* gene, activation and overexpression rather than mutations of this gene have been shown in numerous cancers (breast cancer and ovarian cancer) (*Birnbaum, Sircoulomb & Imbert, 2009*). Stable gene expression levels of the *ERBB2/HER2* gene in the L, [A+B], and D cybrids after treatment suggest that the mtDNA SNP variants do not significantly alter pathways associated with *ERBB2/HER2* gene expression.

## CONCLUSIONS

The efficacy of drug use to treat cancer pathology is determined by the drug's ability to efficiently reduce the effect of or inhibit the disease; however, drug susceptibility has been observed to vary between patients. This variation has been assumed to be a consequence of differences in nDNA, epigenetics, or some other external factors. This study was conducted to assess the consequences of variations within the mtDNA in patients of African, Hispanic, and Asian maternal descent after treatment with the platinum drug cisplatin. The goal of this study was to obtain greater insight into the role of mitochondria, along with its unique genomic information, in drug susceptibility, specifically pathways that promote cisplatin resistance. We used the cybrid model, which are cell lines that contain identical nDNA and media culture conditions but vary only in the specific mtDNA haplogroup profile of each individual. Although nDNA and epigenetic factors may play a role in a patient's response to cisplatin, they would not account for variations in response to cisplatin in the cybrid model as the only variable is the mitochondrial genome. In summary, cybrid cells with mtDNA haplogroups representing different racial/ethnic populations, displayed unique overall results in the various cell assays and gene expression studies (Table 3). If mtDNA played no role in drug susceptibility, the effects of cisplatin would be expected to be similar throughout the study. However, L, [A+B], and D cybrids displayed dissimilar results, indicating that the variations in mtDNA are associated with variations in drug responses. The mechanism for this variation is not yet fully understood, and further studies are needed to understand this process completely.

Future studies will focus on examining similar biological pathways in cybrid cells lines with other mtDNA haplogroups to obtain a greater understanding of drug susceptibility in relation to maternal ethnic descent. Understanding the role of mtDNA haplogroups in drug resistance could potentially improve drug treatment efficiency in vivo , and lead to a greater knowledge of pharmacogenomics as well as lead to a more individualized approach to treatment methods.

## ACKNOWLEDGEMENTS

We wish to thank the subjects who participated in this study.

### Funding

This work was supported by the Discovery Eye Foundation, Polly and Michael Smith, Edith and Roy Carver, and NEI R01 EY0127363 (MCK). Dr. Kenney was supported by an Unrestricted Departmental Grant from Research to Prevent Blindness. We received support from the Institute for Clinical and Translational Science (ICTS) at University of California Irvine. There was no additional external funding received for this study. The funders had no role in study design, data collection and analysis, decision to publish, or preparation of the manuscript.

### Grant Disclosures

The following grant information was disclosed by the authors:
Discovery Eye Foundation, Polly and Michael Smith, Edith and Roy Carver, and NEI: R01 EY0127363.
Research to Prevent Blindness.
Institute for Clinical and Translational Science (ICTS) at University of California Irvine.

### Competing Interests

The authors declare there are no competing interests.

### Author Contributions

- Sina Abedi and Shari R. Atilano conceived and designed the experiments, performed the experiments, analyzed the data, prepared figures and/or tables, authored or reviewed drafts of the paper, and approved the final draft.
- Gregory Yung performed the experiments, analyzed the data, prepared figures and/or tables, authored or reviewed drafts of the paper, and approved the final draft.
- Kunal Thaker and Steven Chang performed the experiments, analyzed the data, authored or reviewed drafts of the paper, and approved the final draft.
- Marilyn Chwa, Nitin Udar, Daniela Bota and M. Cristina Kenney conceived and designed the experiments, authored or reviewed drafts of the paper, and approved the final draft.
- Kevin Schneider performed the experiments, analyzed the data, prepared figures and/or tables, and approved the final draft.

### Human Ethics

The following information was supplied relating to ethical approvals (i.e., approving body and any reference numbers):

All experiments were carried out in accordance with the Institutional Review Board at the University of California, Irvine, (IRB #2003-3131) and were consistent with Federal guidelines.

### Data Availability

Raw and analyzed data available in the Supplementary Files.

## Supplemental Information

Supplemental information for this article can be found online at http://dx.doi.org/10.7717/peerj.9908#supplemental-information.

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
