# Peer review of "Differential effects of cisplatin on cybrid cells with varying mitochondrial DNA haplogroups"

_PeerJ, doi:10.7717/peerj.9908_

## Round 0.1 · original submission · Major Revisions

First, getting reviewers during this time has turned out to be quite challenging and I thank you for your patience and also the reviewers for providing reviews during this time.

The reviewers bring up a number of questions - please review and address these comments for consideration in your resubmission. Questions around the experimental design and statistical analysis were identified as requiring particular clarification.

Reviewer 1 ·

Basic reporting

The article entitled " Differential effects of cisplatin on cybrid cells with varying mitochondrial DNA haplogroups" (article number 45720) submitted by Sina Abedi, et al provided useful information on maternal-origin haplogroup mtDNA plays a role in cisplatin responses. This is a related study from previous studies by the same group (Patel et al. 2019). The authors used transmitochondrial cybrids and NGS techniques and tend to compare the differential effect of mtDNA haplogroups in cisplatin responses. The present study concluded that an individual’s mtDNA background may affect their response to cisplatin treatment, thereby affecting the efficiency and the severity of side effects from the treatment. The underlying molecular mechanisms are still largely unknown.

Experimental design

The comments and suggestions as listed
1. Please change “Method” to “Methods” in Line 51.
2. There is no description on Figures 4 & 5 in the results section. The Figure 4 is not really related to this study, the authors might consider to remove it and describe it in the introduction section and cite the original resource instead. The Figure 5 should be Figure 1 according to the order of the presentation. Thus, the order of Figures should be reformatted.
3. The Figure legends are not properly stated. They should include brief description of experiment procedures (dose, time, conditions…) and the statistical analysis if applied. Conclusion statement is not recommended in the figure legend. In addition, the title of the Figures should be a conclusive statement.
4. According to the information in Materials and Methods section and Table 1, there are only 14 donor subjects in this study (n = 7 for L cybrids, n = 4 for [A+B] cybrids, and n = 3 for D cybrids). Why the number size in different assays is different and greater than the established cybrid lines in Figures 6~9? For example, could it be just because of more sets of data in L haplogroup in Figure 8? It is hard to tell the statistical analysis from the raw data. The same experiments should be done several times and do the statistical analysis afterwards. The authors should describe the statistical analysis in more details and discuss this issue in the discussion section.
5. The authors should provide the rational for picking genes in Tables 2 & 3. The clinical relevance/treatment suggestion??
6. This study provides numerous information of mtDNA haplogroups from racial/ethnic populations correlated with variable susceptibility to cisplatin treatments. Is it cause of effect or only association??
7. The authors should put more efforts on interpretation of the SNPs in the Figs 1-3 in results and discussion sections. Any particular SNP correlated to the observed phenotype in Figures 6~9?

Validity of the findings

From this and others’ studies, It is totally agreed that mtDNA contributes to the susceptibility to cisplatin treatments. However, it is well-known that the nuclear DNA plays important role in cisplatin response. The authors should discuss if there are any role of it in the present study.

Additional comments

The experimental design and statistical analysis are highly recommended to be adjusted.

·

Basic reporting

Please have a look at Major concerns: 6 and Minor concerns: 1 to 4.

Experimental design

Please have a look at Major concerns: 1 to 5.

Validity of the findings

No comment.

Additional comments

Cisplatin, a commonly used chemotherapeutic agent, causes different levels of resistance and side effects for different patients, but the mechanism(s) are presently unknown. It is a good idea to investigate the individual’s mitochondrial (mt) DNA in toxicity of cisplatin. In this manuscript, Dr. Sina Abedi and colleagues used transmitochondrial cytoplasmic hybrids (cybrids) modle to investigate the individual’s mitochondrial (mt) DNA in toxicity of cisplatin in vitro. The study’s conclusion showed that an individual’s mtDNA background may affect their response to cisplatin treatment, thereby affecting the efficiency and the severity of side effects from the treatment. Although the current study is interesting, there are some major concerns that need to be addressed before consideration of publication.

Major concerns:
1. “Materials & Methods – Cybrid Cell Lines Culture” Line 189. “…cells were treated with 100 μL of 40 μM concentration of cisplatin…” Why the author chose cisplatin at a concentration of 40 M? Generally, half of the inhibitory concentration of cisplatin in cancer cells for 48 hours is about 10 M in vitro. It is suggested that the author should inquire and select the experimental concentration according to the blood concentration of cisplatin, or use the gradient concentration of high, medium and low for reference.
2. “Materials & Methods – Cellular Metabolic Activity Measured by Tetrazolium Dye (MTT) Assay” Line 193-205. The principle of MTT method to detect cell activity is related to mitochondrial metabolism. If the survival or viability of cells is to be directly reflected, I suggest the author use Trypan Blue or PI staining cells count method to directly observe the survival or viability of cells.
3. “Materials & Methods – Mitochondrial Membrane Potential (ΔΨm) Assay (JC-1 Assay)” Line 207-217. JC-1 assay is a classical method to detect the mitochondrial membrane potential of cells, but the fluorescence reading by the microplate reader will be affected by the uniformity of cell plated. Therefore, I suggest the author use flow cytometry for quantitative analysis.
4. “Materials & Methods – RNA Isolation, Quantitative-Real Time PCR (qRT-PCR)” Line 232-248. Please provide primer sequence and describe the internal reference gene used.
5. “Materials & Methods – Statistical Analysis” Line 254. In order to better show the degree of data dispersion, it is recommended to use standard deviation instead of standard error.
6. “Results – Growth Rate of Untreated L, A, B, and D Cybrids” Line 264-272. “For example, the untreated-D cybrids (Asian origin) (1.12 ± 0.02, n = 38)…” In Figure 6. The units of the y-coordinate are percentages. Please check the full text carefully and use the same units and data as the chart.

Minor comments:
1. “Introduction” Line 71-150. Background information is too detailed and needs to be drastically reduced.
2. In Figure 6 to 9. The ordinate of all charts should show units.
3. In Figure 6 and 7. The y-coordinate should be “relative cell viability”.
4. In Table 3. HPRT1 and HMBS? Please review the full text carefully and delete the data and text irrelevant to this article.

---

## Round 0.2 · Minor Revisions

Thank you for addressing the reviewer concerns. There is just a single question from reviewer 1 that should be able to be easily addressed before acceptance.

Reviewer 1 ·

Basic reporting

The revised article entitled " Differential effects of cisplatin on cybrid cells with varying mitochondrial DNA haplogroups" (article number 45720) submitted by Sina Abedi, et al provided useful information on maternal-origin haplogroup mtDNA plays a role in cisplatin responses. The authors addressed most of my comments and improved the manuscript to make the content more precise!

Experimental design

The authors addressed most of my comments and modified accordingly.
There is one more question:
I am not quite understand the statistic expression in Fig legends . For example, in Fig 5 “ The untreated-L cybrids (102.4% ±7.0%, SD = 21.0, n = 7) had similar relative cellular metabolic activity when compared to the untreated-D cybrids (111.7% ± 4.3%, SD = 11.3, n = 3, p = 0.311) and the untreated-[A+B] cybrids (95.0% ± 8.5%, SD = 20.9, n = 4, p = 0.512).” What does SD mean in “102.4% ±7.0%, SD = 21.0, n = 7”? Is it 7.0 or 21.0? Same to the rest and other Figures.

Validity of the findings

The authors addressed most of my comments! The authors had discussed in more details about the future application of the current study!

Additional comments

The authors addressed most of my comments and revised the manuscript accordingly.

·

Basic reporting

No comment.

Experimental design

No comment.

Validity of the findings

No comment.

Additional comments

Cisplatin, a commonly used chemotherapeutic agent, causes different levels of resistance and side effects for different patients, but the mechanism(s) are presently unknown. It is a good idea to investigate the individual’s mitochondrial (mt) DNA in toxicity of cisplatin. In this manuscript, Dr. Sina Abedi and colleagues used transmitochondrial cytoplasmic hybrids (cybrids) modle to investigate the individual’s mitochondrial (mt) DNA in toxicity of cisplatin in vitro. The study’s conclusion showed that an individual’s mtDNA background may affect their response to cisplatin treatment, thereby affecting the efficiency and the severity of side effects from the treatment. After the modification, the data quality and language description are improved. Although there are still some deficiencies (sample size is small, and methods is simple) in this study, there are few relevant explorations in this field. This study is a good case worthy of publication and guidance for further research.

---

## Round 0.3 · accepted · Accept

Thank you for addressing the reviewer's question and congratulations again!